# Self-Supervised Goal-Reaching Results in Multi-Agent Cooperation and Exploration

## Abstract

For groups of autonomous agents to achieve a particular goal, they must engage in coordination and long-horizon reasoning. However, designing reward functions to elicit such behavior is challenging. In this paper, we study how self-supervised goal-reaching techniques can be leveraged to enable agents to cooperate. The key idea is that, rather than have agents maximize some scalar reward, agents aim to maximize the likelihood of visiting a certain goal. This problem setting enables human users to specify tasks via a single goal state rather than implementing a complex reward function. While the feedback signal is quite sparse, we will demonstrate that self-supervised goal-reaching techniques enable agents to learn from such feedback. On MARL benchmarks, our proposed method outperforms alternative approaches that have access to the same sparse reward signal as our method. While our method has no *explicit* mechanism for exploration, we observe that self-supervised multi-agent goal-reaching leads to emergent cooperation and exploration in settings where alternative approaches never witness a single successful trial.[1]

## 1 Introduction

Reinforcement learning (RL) has the potential to find novel strategies that solve complex tasks. From controlling fleets of autonomous vehicles to swarms of drones (Cao et al., 2012; Baldazo et al., 2019), cooperative multi-agent RL (Witt et al., 2020; Oliehoek et al., 2016) has the potential to find strategies that are more robust, scalable, and efficient than single-agent strategies. However, finding novel strategies requires that human users do not pigeonhole agents into known solutions with dense rewards. Learning from sparse rewards remains a key open problem in multi-agent RL.

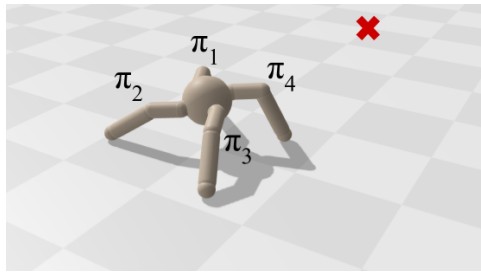

Figure 1: In the **multi-agent goal-reaching** problem, a collection of agents cooperates to maximize the likelihood of visiting a certain state. In this example, four agents coordinate to control an ant-like robot; each agent controls one leg (2 joints/leg). The goal is to coordinate so that the ant moves to a specific position ($\times$). No rewards are given; no distance metrics are required.

In the area of single-agent RL, prior work has demonstrated that agents can learn from sparse rewards — or no reward at all — by leveraging self-supervised techniques (Shelhamer et al., 2016; Achiam et al., 2018; Touati et al., 2022; Ghosh et al., 2021; Eysenbach et al., 2019). Goal-reaching is a canonical example: a human user provides the agent with a goal observation, and the agent attempts to reach that goal as quickly as possible (Kaelbling, 1993; Boyan & Moore, 1994; Dayan & Hinton, 1992; Dietterich, 1998; Sutton, 1995). As the agent receives only a sparse reward upon reaching the goal, it is free to explore and experiment with different strategies for reaching the goal, including strategies that the human designer may not have envisioned. Importantly, prior work in this area has used self-supervised learning to make such sparse reward problems tractable (Ding et al., 2019; Kaelbling, 1993; Lin et al., 2019; Sun et al., 2019).

The main aim of this paper is to study how self-supervised RL techniques can enable groups of agents to cooperate. We will focus on the problem of goal-conditioned multi-agent RL, where a single observation of a desired outcome specifies the task. This problem statement is appealing

---

[1]Project website with code and videos: `https://anonymous.4open.science/r/gcrl_marl`

from a practitioner's perspective (it removes the need for reward design). To study the efficacy of self-supervised learning, we combine insights from recent work on goal reaching in single-agent settings (Eysenbach et al., 2022) and independent learning for MARL (IPPO (Witt et al., 2020)). Our algorithm specifies a shared goal and treats each agent as an independent (self-supervised) learner (with parameter sharing).

The key contribution of this paper is a demonstration that self-supervised goal-reaching is a tractable method for solving complex multi-agent tasks. In experiments, our proposed method achieves substantially higher performance than prior MARL algorithms. All algorithms are given the same sparse feedback. Ours is the only method to get nonzero reward in four environments, and almost triples the win-rate in the fifth. We find that our algorithm explores different coordination strategies over the course of learning, even in settings such as the StarCraft II Multi-Agent Challenge (2s3z, 8m, 6h_v_8z, and 3s_v_5z), where prior methods never observe a single success.

## 2 RELATED WORK

Our work builds on prior work in goal-conditioned RL and independent learning for multi-agent RL (Claus & Boutilier, 1998). In contrast to previous work in unsupervised and sparse-reward MARL, the bulk of which has focused on task-agnostic skill learning (Jiang et al., 2022; Yang et al., 2023) and explicit exploration mechanisms (Na & Moon, 2024; Jeon et al., 2022; Liu et al., 2021), our method learns and explores in a fully end-to-end fashion.

**Independent Learning (IL) in MARL.** Multi-agent reinforcement learning presents a fundamental choice for training: should agents learn together or separately? *Centralized training* approaches (CTDE) have agents share information during training. Methods like COMA and QMIX give agents access to the full environment state and let them share their policies and experiences (Claus & Boutilier, 1998; Foerster et al., 2017; Rashid et al., 2018). After training ends, the agents still execute independently with only partial observations, but they benefit from the shared learning. Our approach builds off of *independent learning* (Tan, 1993) (e.g., IPPO (Witt et al., 2020)), where agents develop their policies without sharing information with each other. Each agent sees only part of the environment and cannot observe what other agents are doing or "thinking" during training. Prior work has shown that Independent PPO outperforms the fully centralized Multi-Agent PPO on complex StarCraft II benchmarks (Yu et al., 2022) and scales better to larger environments (Witt et al., 2020).

**Sparse Reward Methods in MARL.** Prior sparse reward methods in MARL typically use intrinsic motivation and domain-guided search to generate interesting agent behavior in the absence of an explicit reward signal (Jeon et al., 2022; Liu et al., 2023; 2021; Mahajan et al., 2019; Jo et al., 2024; Xu et al., 2023b). MASER, for example, generates sub-goals for *individual* agents from a replay buffer (Jeon et al., 2022). CMAE instead commands *collective* goals by searching for infrequently visited states in projected space to encourage exploration (Liu et al., 2021). LAIES provides intrinsic motivation for causally-meaningful actions, defined using domain knowledge in the cooperative setting (Liu et al., 2023). More recently, methods have maximized diversity between successive joint policies (Xu et al., 2023a) or maintained a set of joint policies that effectively spans large regions of the environment (Xu et al., 2024). Our method will differ by not requiring domain-specific knowledge, subgoals, or explicit intrinsic motivation rewards.

**Goal-Conditioned Reinforcement Learning.** Our work builds on a long line of prior goal-conditioned RL (GCRL) research (Newell et al., 1959; Kaelbling, 1993; Ghosh et al., 2021), wherein a reinforcement learning agent attempts to reach a commanded goal state. While sparse, the goal-conditioned setting is appealing from a user's perspective because it lifts much of the burden of reward function design (Hadfield-Menell et al., 2017; Dulac-Arnold et al., 2019): instead of hand-designing and implementing a reward function, a user simply gives one example of the desired outcome. Prior self-supervised techniques for goal-reaching can tackle long-horizon sparse-reward problems (Lin et al., 2019; Eysenbach et al., 2021; Chen et al., 2021; Eysenbach et al., 2022; Andrychowicz et al., 2017; Liu et al., 2024). Our work extends these self-supervised techniques and observations to the multi-agent setting. Perhaps the most related work is LAGMA (Na & Moon, 2024), which uses goal-conditioned trajectories as an intermediate step when maximizing an extrinsic reward; although our setting will be different as no extrinsic rewards will be provided.

## 3 MULTI-AGENT RL AS A GOAL-REACHING PROBLEM

This section introduces the formal definition of the multi-agent goal-reaching problem after reviewing the standard MARL problem

### 3.1 PRELIMINARIES: MULTI-AGENT RL

We consider a multi-agent RL problem with $n$ agents. At each timestep $t$, each agent receives a local observation $o_t^{(i)}$ and outputs an action $a_t^{(i)}$. Let $s_t^{(i)}$ denote the state of agent $i$, with observations generated according to an agent-specific observation function $o_t^{(i)} \sim O_i(o_t^{(i)} \mid s_t^{(i)})$. The environment transitions according to the stochastic transition function $p(s_{t+1}^{(1:n)} \mid s_t, a_t^{(1:n)})$ with initial state distribution $p_0(s_0^{(1:n)})$. At each timestep, the agents collectively receive a reward $r(o_t^{(1:n)}, a_t^{(1:n)})$. The overall objective is to maximize the expected discounted sum of these rewards:

$$\max_{\pi(a_t^{(i)}|o_t^{(i)})} \mathbb{E}\left[ \sum_{t=0}^{\infty} \gamma^t r(o_t^{(1:n)}, a_t^{(1:n)}) \right]. \tag{1}$$

Following the IPPO paper (Witt et al., 2020), we will treat each agent as an independent policy $\pi(a_t^{(i)} \mid o_t^{(i)})$ that takes actions based on its local observation $o_t^{(i)}$. The parameters are shared across the agents (i.e., each agent has an identical policy). We define a local Q-function for each agent:

$$Q(o_t^{(i)}, a_t^{(i)}) \triangleq \mathbb{E}\left[ \sum_{t'=t}^{\infty} \gamma^{t'-t} r(o_t^{(1:n)}, a_t^{(1:n)}) \mid o_t^{(i)}, a_t^{(i)} \right]. \tag{2}$$

Here, the expectation is taken over the future actions of *all* agents. Our analysis below will make use of the discounted state occupancy measure (Eysenbach et al., 2022; Puterman, 1994; Liu et al., 2024; Ho & Ermon, 2016):

$$\rho_\gamma^\pi(s_f) \triangleq (1-\gamma) \sum_{t=0}^{\infty} \gamma^t p_t^\pi(s_t = s_f), \tag{3}$$

where $p_t^\pi(s_t = s_f)$ represents the probability of the agent being in state $s_f$ at time $t$.

### 3.2 DEFINING THE MULTI-AGENT GOAL-REACHING PROBLEM

We now define the multi-agent goal-reaching problem, building on the Dec-POMDP formalism from prior work (Oliehoek et al., 2016; Witt et al., 2020; Jiang et al., 2022; Na & Moon, 2024) and the GCRL framework (Kaelbling, 1993; Newell et al., 1959; Ghosh et al., 2021). Whereas the Dec-POMDP is typically defined in terms of a reward function, we will omit the rewards and instead include a space of goals.

We start by introducing the goals. Let $\mathcal{G}$ be a space of goals and let $m_g^{(1:N)} : \mathcal{O}^{(1:N)} \to \mathcal{G}$ be a mapping from the observation space of all agents to the collective goal space. Goals are defined in this way because the objective is typically not to reach a particular state, but rather to reach *any* state that satisfies a desired property (e.g., any state where the agents have successfully shot a basketball into a hoop). Let $p_g(g)$ denote the distribution over goals used for data collection and evaluation. Many of our experiments will use $p_g(g) = \delta(g = g^*)$, a Dirac distribution at one particular goal of interest (e.g., when all opponents have been defeated). Crucially, this choice of goal distribution eliminates the need to pre-define or adapt a goal curriculum, as done in prior work (Liu et al., 2021; Jeon et al., 2022; Na & Moon, 2024).

We command the agents to reach an element of the goal space and consider the objective achieved if the agents occupy a state that corresponds to the goal. As in the GCRL framework, the optimization objective is to maximize the probability of reaching the goal state, where the goal is a function of observations $m_g^{(1:N)}(o^{(1:N)}) = g$. Note that such mappings from full observations to relevant subsets of the observation for the goal are standard in problem settings from various prior works (Lin et al., 2019; Bortkiewicz et al., 2024; Liu et al., 2021). We explore relaxing this setting in Appendix D.2.

To cast the goal-reaching setting as a reinforcement learning problem, we define the reward as

$$r(o_t^{(1:N)}, a_t^{(1:N)}) = \begin{cases} 1 & \text{if } m_g^{(1:N)}(o_t^{(1:N)}) = g \\ 0 & \text{otherwise} \end{cases}. \tag{4}$$

As it is currently stated, this reward definition does not make sense in settings with continuous states, as hitting the goal would be a measure-0 event. Thus, for generality, we define the reward function as the likelihood of hitting the goal at the *next* time step:

$$r(o_t^{(1:N)}, a_t^{(1:N)}) = p(m_g(o_{t+1}^{(1:N)}) = g \mid o_t^{(1:N)}, a_t^{(1:N)}). \tag{5}$$

These two reward functions are equivalent in expectation and thus result in equivalent optimization objectives (Eq. 1) (Eysenbach et al., 2022). In summary, the overall objective is:

$$\max_{\pi(a^{(i)}|o^{(i)},g)} \mathbb{E}_{p_g(g), \pi(\tau^{(1:N)}|g)} \left[ \sum_{t=0}^{\infty} \gamma^t r(o_t^{(1:N)}, a_t^{(1:N)}) \right] = \max_{\pi(a^{(i)}|o^{(i)},g)} \mathbb{E}_{p_g(g)}[\rho_\gamma^\pi(g)] \tag{6}$$

where $\tau^{(i)}$ is the sequence of observations and actions seen by agent $i$, $\pi(\tau^{(i)}|g)$ is the probability of sampling such a sequence given policy $\pi$ and goal $g$, and $\rho_\gamma^\pi(g)$ is the discounted state occupancy measure (Eq. 3). Intuitively, this objective corresponds to maximizing the time spent in the commanded goal $g$. We note that using such a sparse, task-specific reward function is not new in MARL, and has been previously considered in a non-goal-conditioned setting (Liu et al., 2021; Xu et al., 2023b; 2024). However, the approach for solving this problem, which we present in the next section, differs from prior work by avoiding additional goal search, goal sampling, or use of task-specific goal knowledge (Jeon et al., 2022; Liu et al., 2023; Xu et al., 2023b).

## 4 METHOD: INDEPENDENT CRL

We introduce an actor-critic algorithm for multi-agent goal reaching. Following prior work (Witt et al., 2020; Tan, 1993), we will address the multi-agent setting by learning decentralized policies and Q-functions.

**Critic objective.** We will use a variant of contrastive RL (CRL) (Eysenbach et al., 2022) to learn the Q-function, which is defined using a sparse goal-reaching reward (Eq. 5). CRL is a goal-conditioned RL (GCRL) method that uses a temporal contrastive objective to learn representations without requiring an external reward signal. At a high level, the aim of this method is to learn representations from data which capture the relatedness of state-action pairs and goals in the given multi-agent environment. The core of this method consists of two learnable encoders $\phi(o, a)$ and $\psi(g)$ that capture control-relevant temporal correlations between $(o, a)$ and $g$. At optimum, the learned representations parameterize the conditional discounted state occupancy measure $\rho^\pi(g \mid o, a)$ (Eq. 6); this is our critic in the goal-conditioned setting. Intuitively, these representations should enable agents to perform exploration directed toward a goal even when that goal has never been achieved. Note that for environments with discrete actions, we use a variant of the Gumbel-Softmax trick (see Appendix C). Formally, the (scaled) Q-function can be modeled as the exponential of the distance between two representations. Formally, we define the distance function as:

$$f_{\phi,\psi}(o_t^{(i)}, a_t^{(i)}, g) = -\|\phi(o_t^{(i)}, a_t^{(i)}) - \psi(g)\|_2.$$

We learn these representations by optimizing the symmetric InfoNCE loss (Bortkiewicz et al., 2024; Radford et al., 2021), which reformulates the problem of learning relative probabilities as a classification task between different distributions. The symmetric InfoNCE loss (Eq. 7) relies on a batch of samples $\mathcal{B}$ with pairs $(o_i, a_i), g_i$ as positive examples and pairs $(o_i, a_i), g_j$ and $(o_j, a_j), g_i$ as negative examples, where $g_i$ are future states encountered after $(o_i, a_i)$. Upon convergence, the classifier $C_{\phi,\psi}((o, a), g) = \log \left( \frac{\exp(f_{\phi,\psi}(o,a,g))}{\sum \exp(f_{\phi,\psi}(o,a,g))} \right)$ learns temporal relationships between the underlying $(o, a)$ and $g$. The full symmetric InfoNCE loss is as follows:

$$\min_{\phi,\psi} \mathbb{E}_{\mathcal{B}} \left[ -\sum_{i=1}^{|\mathcal{B}|} \log \left( \frac{e^{f_{\phi,\psi}(o_i,a_i,g_i)}}{\sum_{j=1}^{K} e^{f_{\phi,\psi}(o_i,a_i,g_j)}} \right) - \sum_{i=1}^{|\mathcal{B}|} \log \left( \frac{e^{f_{\phi,\psi}(o_i,a_i,g_i)}}{\sum_{j=1}^{K} e^{f_{\phi,\psi}(o_j,a_j,g_i)}} \right) + 0.01 \cdot R(\phi,\psi) \right], \tag{7}$$

$$\text{where} \quad R(\phi,\psi) \triangleq \log \left( \sum_{j=1}^{K} e^{f_{\phi,\psi}(o_i,a_i,g_j)} \right) + \log \left( \sum_{j=1}^{K} e^{f_{\phi,\psi}(o_j,a_j,g_i)} \right).$$

---

**Algorithm 1 Independent CRL** is an actor-critic algorithm for multi-agent goal reaching.

---

Initialize policy $\pi_\theta(a_t^{(i)} \mid o_t^{(i)}, g)$, decentralized critic $f_{\phi,\psi}(s_t^{(i)}, a_t^{(i)}, g) = \|\phi(o_t^{(i)}, a_t^{(i)}) - \psi(g)\|_2$, and replay buffer $\mathcal{B}$.

**while** not converged **do**

    Sample goal $g \sim p_g(g)$ from the commanded goal distribution $p_g(g) = \delta_g(g)$.

    Collect episode using (independent) policies $\pi(a_t^{(i)} \mid o_t^{(i)}, g)$.

    Store episode $\{o_0^{(1:n)}, a_0^{(1:n)}, o_1^{(1:n)}, a_1^{(1:n)}, \cdots\}$ in buffer $\mathcal{B}$.

    Sample observations $o_t^{(i)}$, actions $a_t^{(i)}$, and achieved future goals $g^{(i)}$ from $\mathcal{B}$.

    Update critic $f_{\phi,\psi}$ with temporal contrastive learning (Eq. 7).

    Update policy $\pi_\theta(a_t^{(i)} \mid o_t^{(i)}, g_t^{(i)})$ to maximize the critic (Eq. 9).

**return** policy $\pi_\theta(a_t^{(i)} \mid s_t^{(i)}, g)$.

---

The multi-agent setting involves multiple interacting agents' states, observations, actions, and goals, thus admitting many different sampling schemes in the InfoNCE loss. Here, we choose to sample positive observation-action pairs and goal examples from the experiences of *the same* randomly-chosen agent. Negative samples are drawn uniformly at random over all agents' goals, irrespective of identity. In other words, the trained critic prioritizes learning features that help agents distinguish goals encountered in their *own* futures from the randomly-drawn futures of all agents.

Mathematically, a classifier $C_{\phi,\psi}$ trained over these distributions converges to an averaged Q-function over agents (see Eq. 2): $C_{\phi,\psi}^*((o, a), g) \propto \rho_\gamma^{\text{mix}}(m_g(o_{t+}) = g \mid o_t, a_t)$, where $\rho_\gamma^{\text{mix}}$ denotes the $\gamma$-discounted state occupancy measure ("local critic") averaged over agents $i$ ("mixed critic"):

$$\rho_\gamma^{\text{mix}}(m_g(o_{t+}) = g \mid o_t, a_t) \triangleq \frac{1}{N} \sum_i \rho_\gamma^{(i)}(m_g(o_{t+}^{(i)}) = g \mid o_t^{(i)}, a_t^{(i)}).$$

Thus, correctly learning this classifier enables any individual agent to choose actions that maximize the *mixed* probability of occupying a goal state, an approximation to the full goal-reaching objective (Eq. 6). Importantly, we assume that the overall goal $g$ can be approximated as a function of local observations $o^{(i)}$. While not true in the general MARL setting, we found that this assumption leads to strong empirical performance in complex collaborative benchmarks.

**Actor objective.** We use a neural network policy $\pi_\theta(a_t^{(i)} \mid o_t^{(i)}, g)$ that takes *individual* observations $o_t^i$ and goal as input. We assume the agents are homogeneous, so the same policy is used for modeling all agents (i.e., we employ parameter sharing). Non-homogeneous tasks can be made homogeneous by including the agent index or type as part of the observation space.

Following single-agent CRL methods (Eysenbach et al., 2022), we train the policy by maximizing the (marginal-weighted) expected critic over states and goals sampled from the replay buffer:

$$\max_\pi \mathbb{E}_{\substack{(o_t^{(i)}, g^{(i)}) \sim \mathcal{B} \\ a_t^{(i)} \sim \pi(a_t^{(i)} \mid o_t^{(i)}, g^{(i)})}} [-\|\phi(o_t^{(i)}, a_t^{(i)}) - \psi(g^{(i)})\|_2]. \tag{8}$$

The policy should pick actions such that the current state-action pair is close to the goal state in representation space. Once again, observations and future goals are sampled from the trajectories of the *same* randomly-chosen agent, an approximation of the full MARL observations and actions over all agents. We show that maximizing this objective is equivalent to maximizing a lower bound on the full contrastive actor objective over all agents; further details are in Appendix E.

### 4.1 Algorithm Summary

We now summarize our complete algorithm, Independent CRL, and provide pseudocode in Alg. 1. Independent CRL works in the online setting, alternating between collecting data and updating the actor and the critic. In order to update the actor and critic, a batch of state-action pairs are sampled from the replay buffer. We implement Independent CRL on top of JaxGCRL (Bortkiewicz et al., 2024). Code to reproduce our experiments is available in a code repository.

## 5 Experiments

In this section, we aim to answer the following questions:

**(Q1)** Is efficient exploration possible in long-horizon, sparse-reward tasks?

**(Q2)** How does the performance of Independent CRL compare to hierarchical approaches?

**(Q3)** How does exploration emerge in our method to solve these sparse learning tasks?

**(Q4)** How does Independent CRL perform on continuous control tasks, which introduce complexity in manipulation?

**(Q5)** Does reframing a goal-reaching task as multi-agent make it harder or easier?

We present all results with $\pm 1\sigma$ error bars and average smoothing (to display 200 data points). The experiments (5 seeds per baseline per experiment) required 0.5-3 hours (ICRL, IPPO, and MAPPO) and 16 hours (MASER) per seed on a Tesla V100 GPU (32 GB). All experiments were run on an internal cluster. See Appendix A for a summary of all experiments and Appendix B for additional experimental details.

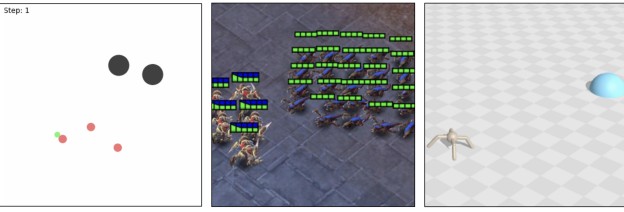

Figure 2: **Multi-agent environments.** MPE Tag (Rutherford et al., 2024), SMAX (SMAC) (Rutherford et al., 2024; Samvelyan et al., 2019), and multi-agent Ant (Rutherford et al., 2024).

## 5.1 LONG-HORIZON COOPERATION

While the agents perform well on didactic multi-particle (MPE) Tag environments where they observe the goal-state frequently (see Appendix D.1), it is intuitively unclear whether this method should work on long-horizon tasks where the goal-state is only observed once after many (e.g., 50-100) steps. Can agents explore effectively for long periods without any feedback?

To answer this question, we evaluate the performance of ICRL on the StarCraft Multi-Agent Challenge (SMAC), a common benchmark in prior MARL work that pits a team of units against an enemy team (Samvelyan et al., 2019). For our experiments, we use SMAX, a JAX implementation of a SMAC-like environment (Rutherford et al., 2024). We test five classic SMAC environments: `3m`, `2s3z`, `6h_v_8z`, `8m`, `3s_v_5z`, as well as SMACv2 environments featuring random position and unit-types. We compare Independent CRL with the multi-agent baselines IPPO and MAPPO, both of which have been previously found to be effective on a variety of SMAC environments (Yu et al., 2022).

We frame the goal as: reduce the sum of enemy healths to zero. Mathematically, this means that $m_g(o_t^{(i)})$ is the sum of the enemy healths and the goal $g$ is the scalar 0. If an enemy is not observed, its full health is added to the sum. To ensure fair comparison with our goal-conditioned method, we provide the reward-driven baselines with a sparse reward (namely, a reward of 1 is given upon winning a battle).

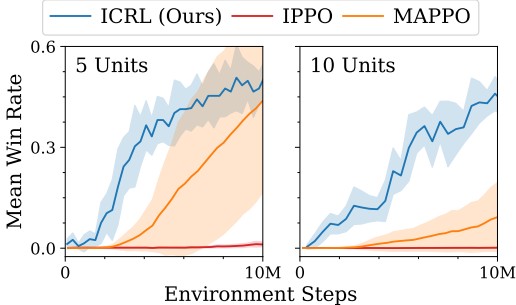

We empirically observe (Figure 4) that results are very inconsistent for the non-goal conditioned methods on the SMAC environments, which often fail to see a single success throughout the entirety of a training run. Sometimes, as we see for MAPPO on the `3m` environment, lucky successes may lead to enough signal to

Figure 3: We compare ICRL (ours) to IPPO and MAPPO on the randomized 5-agent and 10-agent SMACv2 environments. Our approach learns faster than both baselines, achieving higher asymptotic returns than IPPO. IPPO's win rate is effectively zero.

learn interesting behavior; however, in the absence of dense rewards, we find that their exploration is very inconsistent. On the other hand, **we find that Independent CRL is consistently able to identify successful policies early on in the training process**, suggesting that the use of goal-conditioned representation learning enables the method to perform effective multi-agent exploration with minimal supervision (a finding we will later validate qualitatively in Sec. 5.3).

As shown in Figure 3, on the 5-agent and 10-agent SMACv2 environments, we again find that **our method can consistently learn how to coordinate to defeat the enemy, learning more quickly than both IPPO and MAPPO**. While, for the 5-agent environment, MAPPO converges to a more

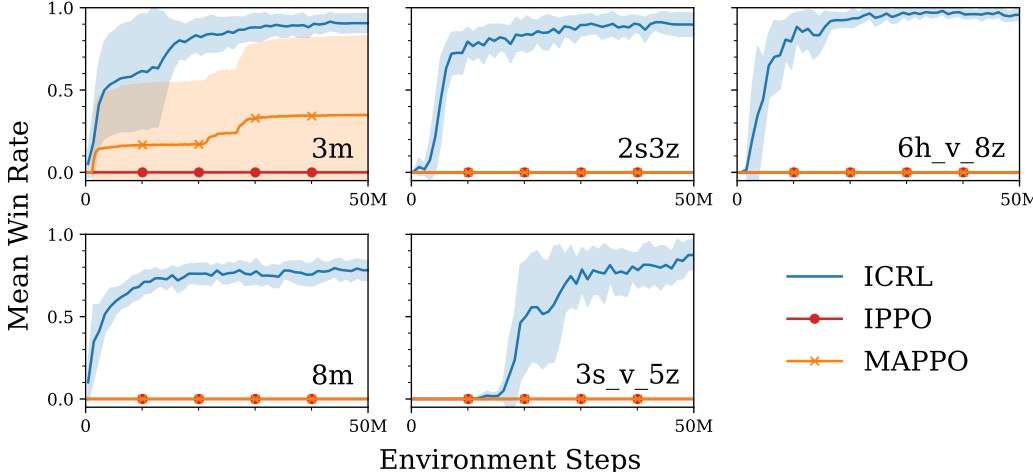

Figure 4: **Efficient learning on the StarCraft Multi-Agent Challenge (SMAX).** We compare ICRL (our method) to IPPO and MAPPO on five settings from the SMAX benchmark. On the 3m setting, our method achieves a win rate that is $\sim 3\times$ higher than MAPPO, while the IPPO baseline has a win rate of zero. On the 2s3z, 6h_v_8z, 8m, and 3s_v_5z settings, only our method achieves a non-zero win rate.

successful policy on average, ICRL achieves a higher overall win-rate for the more challenging 10-agent version. We suspect that the inherent stochasticity present in this environment may naturally encourage agents to explore more widely, leading to the reward-based baselines performing better on these tasks than in the standard SMAC task. A statistical significance test located in Appendix D.3 shows that the probability of improvement for ICRL compared to MAPPO is on average 94%.

## 5.2 ARE HIERARCHICAL METHODS NEEDED FOR HIGH PERFORMANCE IN SPARSE-REWARDS?

A long line of prior work suggests that hierarchical approaches–methods that decompose long-horizon tasks into a sequence of shorter/easier problems–are crucial for solving long-horizon tasks, such as those in SMAC (Jeon et al., 2022; Liu et al., 2021; Mahajan et al., 2019; Na & Moon, 2024). Indeed, prior work (Jeon et al., 2022) has found such designs highly effective in the multi-agent setting. Our next experiment studies whether these design ingredients are required. To do this, we compare ICRL–which does not include any subgoals or intrinsic rewards–to MASER, a SOTA method that tackles the challenge of sparse rewards by generating and assigning subgoals.

For fair comparison, both methods receive the same sparse rewards (+1 if win, 0 otherwise); this is a sparser reward than presented in the original paper, which gives individual enemy reward bonuses and penalties for ally health loss. Again, for fair comparison, we use feedforward networks for both methods. As shown in Fig. 5, **ICRL achieves earlier wins with fewer samples, as well as an asymptotically higher win rate.**

## 5.3 EMERGENT EXPLORATION

As shown in previous experiments, our method performs well in long-horizon tasks where there is no signal until the end of the episode. For this to occur, agents must explore effectively in

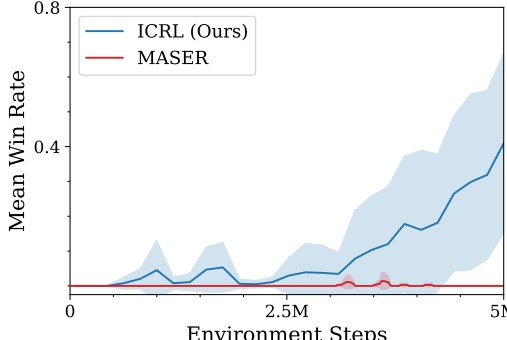

Figure 5: ICRL (Ours) vs. MASER on SMAC (2s3z), using sparse rewards. Our method achieves 60% win-rate by step 5M while MASER's win-rate is negligible.

the interim before receiving any signal about which methods actually lead to the goal. How does exploration occur before the algorithm has even observed a single success? By visualizing learned policies at various points along training, we can see how exploration emerges in training and what, if any, unique skills are learned.

Previous work discussed emergent self-directed exploration in the single-agent setting, showing that giving a single goal leads to effective exploration (Liu et al., 2024). That paper demonstrates contrastive representations are critical to learning useful skills before a single success is observed (a

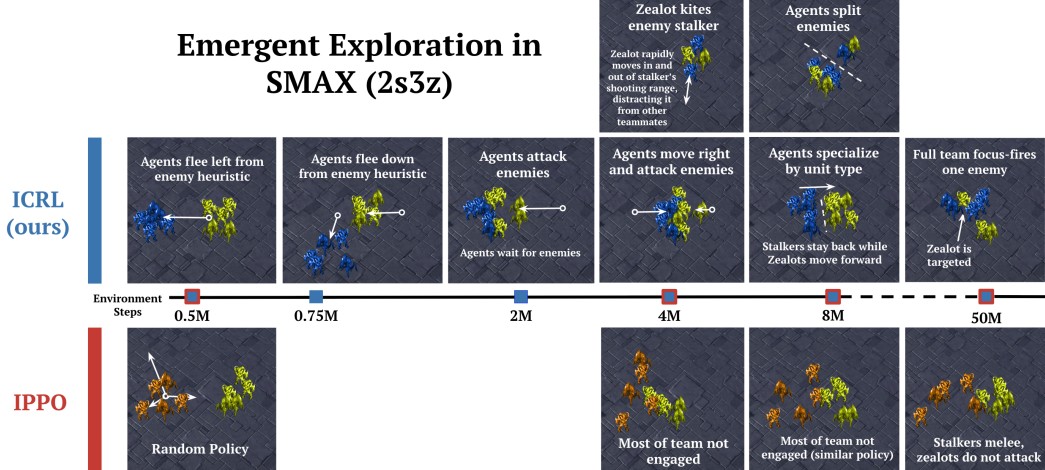

Figure 6: **How does ICRL learn to play StarCraft?** We visualize the exploration strategies of *(Top Row)* ICRL (our method) and *(Bottom Row)* IPPO on the SMAX (2s3z) environment over 50 million training environment steps. We observe that the ICRL algorithm explores different coordination strategies over the course of learning.

monolithic critic does not show directed exploration) (Liu et al., 2024). By extension to the multi-agent setting, we suspect that commanding a single goal to our method also enables such directed exploration. We empirically find that the learned curriculum of skills from ICRL surpasses subgoal-generating MASER, mirroring results in the single-agent setting (Liu et al., 2024). We conclude, similar to previous work (Liu et al., 2024), that emergent exploration occurs under the following circumstances/conditions: *(1)* environmental dynamics are learned in contrastive representations and *(2)* a single long-horizon goal is commanded for CRL (rather than many human-picked subgoals).

We use the 2s3z environment (Figure 6) to illustrate emergent exploration. **Our method learns basic skills (e.g., movement, attacking) early in training and slowly learns more advanced strategies, often well before observing a single success.** Initially, agents flee from the enemy units and hit the boundary, where they are eliminated. Later, agents learn to stay still and shoot the incoming enemy. By 2 million environment steps, before the agents have seen the goal-state even once, agents have explored various unique strategies. At 4 million steps, we observe common StarCraft unit micromanagement techniques: units learn unique skills such as "kiting" and focus-fire (Rutherford et al., 2024). At 8 million steps, agents learn more optimized flocking behaviors and learn to specialize by their unit type (we explore specialization more in Appendix D.4). Despite training with shared parameters, the policy network learns unique behaviors (for example, ranged attacks for the stalker or closer melee for the zealot unit). By the end of training, the agents not only retain the most successful skills, but also sync movements and focus-fire all at the same target. This is in contrast to behaviors learned by non-goal conditioned methods such as IPPO, which generally cannot perform directed exploration without much signal. We observe that IPPO policies remain qualitatively similar and behave essentially randomly in benchmark training runs.

### 5.4 TASKS WITH CONTINUOUS ACTIONS

The tasks we have investigated so far have featured straightforward control mechanisms. Many real-world tasks, however, have an orthogonal dimension of complexity: manipulation of the agent in the physical world. Multi-Agent MuJoCo factorizes classic control tasks such as ant and half-cheetah, giving each agent control over only a subset of the available joints and observability of only the nearest other agents (Peng et al., 2021). Multi-Agent BRAX implements five of the Multi-Agent MuJoCo tasks (Rutherford et al., 2024); we will use the Ant task (see Fig. 1). We observe our that method performs well on this task and in Appendix 5.5 we explore how reframing even single-agent tasks as multi-agent increases performance.

We define the goal as controlling the Ant so that its center of mass is at the desired goal. We define $m_g(o_t^{(i)})$ to extract the Ant's $(x, y)$ position and sample goals $g \in \mathrm{R}^2$ uniformly on a disk of radius 10m. The reason we chose this specific distribution is that it is used by the JaxGCRL benchmark (Bortkiewicz et al., 2024) as the default goal distribution for the Ant task, making for a challenging but reasonable goal so that agents have to learn a non-trivial walking policy. For evaluation purposes, a success is measured as a time-step (out of a total 1000) when the agent is within 0.5 meters of the goal.

While our method does not require rewards (it simply maximizes the likelihood of an event), the baselines use this success metric as a sparse reward. The results, shown in Figure 7, show that **our method is able to consistently reach the commanded goal on the sparse reward setting**. IPPO struggles to get non-zero reward, likely because the sparsity of the task makes exploration challenging.

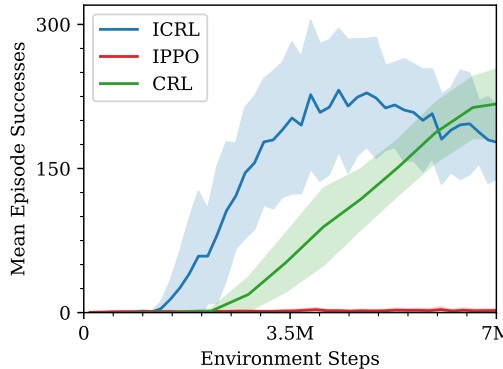

### 5.5 REFRAMING SINGLE-AGENT PROBLEMS AS MULTI-AGENT PROBLEMS

While multi-agent reinforcement learning (MARL) is typically viewed as an extension of single-agent RL—with multi-agent tasks considered strictly harder versions of their single-agent counterparts—MARL can also be understood as making independence assumptions that create more structured search spaces. These independence structures, which enable more data-efficient learning and better generalization in

Figure 7: **Can ICRL solve continuous control tasks?** We compare ICRL (ours) to IPPO for controlling a 4-legged robot: each leg's joints are controlled by a separate agent (Rutherford et al., 2024). IPPO makes no progress, perhaps because the sparse reward signal makes exploration challenging. Single-agent CRL is also compared, showing that casting even a single-agent problem as multi-agent allows the agent to learn faster.

other areas of machine learning (Koller & Friedman, 2009), motivate us to turn the standard MARL problem on its head: Does treating a single-agent RL problem as a multi-agent problem allow us to learn more efficiently? We compare our method versus contrastive RL (Eysenbach et al., 2022) on the same ant task, summarized in Figure 7. This helps answer: Does multi-agent cooperation pose another challenge on top of goal-conditioned policy learning?

Surprisingly, we observe that Independent CRL initially outperforms CRL. **While it may seem that the factored and partially-observed multi-agent version of the task is necessarily more difficult, we find this is not the case.** It appears that factoring the robot into various independently-controlled agents actually reduces the hypothesis space: instead of searching over all policies involving eight joints in the four legs, each agent can focus on controlling just two joints in its corresponding leg. This would imply trading off low variance (leading to faster learning and convergence) with higher bias (leading to worse overall performance). We empirically observe this exact trade-off: Independent CRL starts achieving success in 1 million steps (versus 2 million for CRL) but levels off while CRL continues improving.

## 6 CONCLUSION

In this paper, we have studied the problem of goal-reaching in multi-agent settings, a problem statement designed to lift the human burden of reward engineering. While this problem entails very sparse feedback, our experiments demonstrate that the proposed method can make progress on solving these tasks. We do not claim that our method is anywhere near the maximum possible performance, but rather offer these experiments as evidence that multi-agent goal-reaching is a tractable problem statement. Our hope is to encourage future work to study this appealing problem setting.

One intriguing observation, which we intend to study in future work, is that our independent CRL method appears to do effective exploration despite not having an *explicit* exploration mechanism. The fact that some prior methods *never* make any learning progress suggests that the good performance of our method cannot solely be explained by effective learning from successes, but also must partially be explained by a capacity to explore. This observation is in line with prior work in the single-agent setting (Liu et al., 2024), yet, to the best of our knowledge, there is still no theoretical explanation for why these self-supervised goal-reaching algorithms exhibit "emergent" exploration.

**Limitations.** Although formulating a problem as goal-reaching allows the task to be specified more easily via a single goal rather than a reward function, it may not always be clear how to specify certain tasks as goals. It is also possible for there to be different choices of $\mathcal{G}$ and $m_g$ that may be equally valid in expressing the goal, yet result in slightly different learning behaviors. See discussion in Appendix D.2.

**Reproducibility Statement.** All code needed to reproduce our experiments is available in our anonymous code repository: https://anonymous.4open.science/r/gcrl_marl. Additional proofs for theoretical results about our method are available in Appendix E.

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

**LLM Usage Statement**   We do not believe that we used LLMs significantly to the extent that they could be regarded as contributor. We used LLMs for the use of: generating visually-appealing figures, advice on phrasing for writing of a few sentences, help debugging code, research into finding three related papers.

# A   MAIN EXPERIMENTAL RESULTS

**Restatement** (Experimental Setup). *All experiments use sparse 0/1 rewards: +1 when in the goal state, 0 otherwise. For SMAX environments, the goal is to reduce enemy health to zero. For continuous control, the goals are spatial positions. We compare against IPPO, MAPPO, and MASER baselines using identical sparse reward signals.*

Table 1 summarizes the main experimental results:

| Environment (Metric) | Task | Approach | Result |
|---|---|---|---|
| **MPE Tag** (Mean Episode Return) | 3 Agents | **ICRL (Ours)** | **4704.62 ± 850.18** |
| | | IPPO (Witt et al., 2020) | 3643.21 ± 82.01 |
| | 6 Agents | **ICRL (Ours)** | **16293.59 ± 2204.55** |
| | | IPPO | 5158.75 ± 1759.75 |
| **Multi-Agent Control** (Mean Success Rate) | Ant | **ICRL (Ours)** | **270.63 ± 30.02** |
| | | IPPO | 6.94 ± 8.94 |
| **StarCraft Multi-Agent Challenge (SMAX)** (Mean Win Rate) | 3m | **ICRL (Ours)** | **0.94 ± 0.06** |
| | | IPPO | 0.00 ± 0.00 |
| | | MAPPO (Yu et al., 2022) | 0.36 ± 0.49 |
| | 2s3z | **ICRL (Ours)** | **0.95 ± 0.02** |
| | | IPPO | 0.00 ± 0.00 |
| | | MAPPO | 0.00 ± 0.00 |
| | | MASER (Jeon et al., 2022) | 0.01 ± 0.02 |
| | 6h_v_8z | **ICRL (Ours)** | **1.00 ± 0.00** |
| | | IPPO | 0.00 ± 0.00 |
| | | MAPPO | 0.00 ± 0.00 |
| | 8m | **ICRL (Ours)** | **0.84 ± 0.07** |
| | | IPPO | 0.00 ± 0.00 |
| | | MAPPO | 0.00 ± 0.00 |
| | 3s_v_5z | **ICRL (Ours)** | **0.95 ± 0.03** |
| | | IPPO | 0.00 ± 0.00 |
| | | MAPPO | 0.00 ± 0.00 |

Table 1: Maximum performance across environments. We report episode returns for MPE Tag, success rate for Multi-Agent Control, and win-rate for SMAX tasks. All values are reported as the maximum result $\pm 1\sigma$ at the timestep the maximum result is achieved.

## A.1   ADDRESSING LOW BASELINE PERFORMANCE ON SPARSE REWARDS

To ensure fair comparisons in our goal-conditioned setting (where an agent only gets reward signal when in the goal state), we give all methods access to the same sparse rewards: $+1$ if in the goal state (e.g., win state in SMAX or the goal position in MABRAX Ant) and $0$ otherwise.

The goal-conditioned 0/1 reward specification is a natural choice for any task-specific objective, where the successful completion of the task is a function of agent observations. Notably, the goal-conditioned reward does not need any domain-specific knowledge beyond the specification of the task itself. Unlike LAIES (Liu et al., 2023), we do not need to specify external states in fully cooperative settings, which would require domain-specific knowledge for different tasks. Furthermore, this 0/1 reward specification is not new and has been previously used to benchmark methods: other papers such as CMAE (Liu et al., 2021) and LAIES (Liu et al., 2023) also use such a setting for their method comparisons, and find similarly low performance in their tested baselines (though the compared methods are different).

To address the concern that the IPPO and MAPPO baselines are not representative of the current SOTA in sparse reward settings (such as the goal-conditioned setting), we directly compare our method to MASER (Jeon et al., 2022), a MARL algorithm designed for sparse reward that has shown superior performance to state-of-the-art MARL algorithms including QMIX (Rashid et al., 2018), MAVEN (Mahajan et al., 2019), and COMA (Foerster et al., 2017) without further signal from expert domain knowledge. ICRL outperforms MASER in the 0/1 reward setting (Figure 5). We believe at least one of the reasons for ICRL's high relative performance on such sparse-signal environments is the method's ability to explore effectively. This emergent exploration is discussed further in Section 5.3.

## B    EXPERIMENTAL DETAILS

> **Restatement** (Independent CRL Method)**.** *Our method treats each agent as an independent contrastive learner with shared parameters. The critic learns representations $\phi(o, a)$ and $\psi(g)$ using the symmetric InfoNCE loss, while the actor maximizes $\mathbb{E}[-\|\phi(o_t^{(i)}, a_t^{(i)}) - \psi(g)\|_2]$ to output actions that are close to the goal in representation space.*

Code and hyperparameters for reproducing all experiments can be found in a code repository.[2] We highlight some key hyperparameters in Table 2.

| Hyperparameter | Value |
|---|---|
| Total Environment Steps | 50,000,000 |
| # Epochs | 500 |
| # Environments | 256 (64) |
| # Eval Environments | 64 |
| Actor LR | 3e-4 |
| Critic LR | 3e-4 |
| Alpha LR | 3e-4 |
| Batch Size | 256 (64) |
| Gamma | 0.99 |
| LogSumExp Penalty Coefficient | 0.1 |
| Max Replay Size | 5,000 |
| Min Replay Size | 1,000 |
| Unroll Length | 62 |

Table 2: Independent CRL Hyperparameter Values for the MPE Tag, MABRAX, and SMAX environments. Note for the 6h_v_8z SMAX environments, we needed to reduce environments and batch size (listed in parentheses) to avoid out-of-memory errors.

We found our method to be fairly robust: we did not perform any hyperparameter tuning across the various environments. One exception was for the larger SMAX environments, where we reduced the

---

[2]https://anonymous.4open.science/r/gcrl_marl

number of environments and batch size to avoid out-of-memory errors. For all experiments, we test all algorithms without RNNs. For our SMAX experiments, we use the available action mask for all algorithms.

## C HANDLING DISCRETE ACTIONS

In our experiments, we use both environments with continuous actions and environments with discrete actions. For the discrete action tasks, we train the policy network with the Straight-Through Gumbel-Softmax trick (Jang et al., 2016; Maddison et al., 2016) for backpropagating gradients through the sampling of the discrete action. We make an unconventional choice by parameterizing our critic as a function of the soft actor output rather than the discrete action itself; empirically, we found that this slightly speeds up learning for our algorithm. By using the soft outputs in both the forward and backward pass during actor/critic optimization, we avoid approximating any gradients. Please refer to the code block below for further clarification.

```
hard_actions = jax.nn.one_hot(jax.nn.argmax(logits), num_actions)
soft_actions = jax.nn.softmax(logits, axis=-1)

# CRITIC LOSS
loss = critic_loss(obs, soft_actions, achieved_goals)      # ours
# loss = critic_loss(obs, hard_actions, achieved_goals)    # conventional

# ACTOR LOSS
loss = f(obs, soft_actions, achieved_goals)                # ours
# loss = f(obs, (hard_actions - soft_actions).detach() +
#         soft_actions, achieved_goals)                    # conventional
```

## D ADDITIONAL EXPERIMENTS

### D.1 DIDACTIC EXPERIMENT: TEAMWORK ON MPE TAG

Our first experiment aims to study whether our algorithm ICRL works at all. To test this, we choose a simple task where prior methods are known to work.

In the MPE Tag environment, predator agents must collide with (or "tag") a faster prey. The environment is partially observed: if agents or landmarks are outside a fixed view radius, their observation gets masked with a placeholder value. We use the FACMAC (Factored Multi-Agent Centralized Policy Gradients) variant, where the prey uses a simple heuristic policy, so the environment is a fully cooperative Dec-POMDP (Rutherford et al., 2024; Peng et al., 2021). We frame the goal as: set the distance of the nearest (observed) predator to the prey to zero. Mathematically, we set $m_g(o_t^{(i)})$ as the distance between the closest predator and prey, and estimate the goal $g$ equal to the scalar $0$. If the prey is not visible, this is set to an arbitrarily large value. The results are summarized in Figure 8.

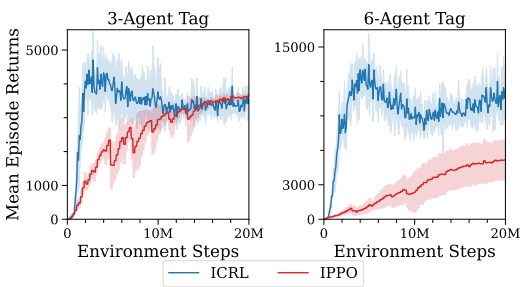

Figure 8: **"Tag" as a goal-reaching problem.** We compare ICRL (ours) to IPPO on the Multi-Particle Agent Tag FACMAC Environment, including both the 3-agent setting *(Left)* and the 6-agent setting *(Right)*. We observe that our method learns faster in both settings and reaches higher asymptotic returns in the 6-agent setting.

Our results demonstrate that Independent CRL consistently performs well against baseline methods. On the MPE Tag task, **Independent CRL matched or beat the performance of IPPO both qualitatively (via interesting formations and strategies) and quantitatively (gaining more rewards at**

**its peak**). While Independent CRL matched performance on the 3-agent environment, it exceeded the IPPO baseline on 6 agents, converging to an effective policy much more quickly.

## D.2 ROBUSTNESS TO NOT SPECIFYING GOAL-SPACE $\mathcal{G}$ AND GOAL-MAPPING $m_g$

> **Restatement** (Multi-Agent Goal-Conditioned RL Problem). *We consider a multi-agent RL problem where agents cooperate to reach a commanded goal state $g \in \mathcal{G}$, with mapping $m_g : \mathcal{O} \to \mathcal{G}$ from observations to goals. The goal-conditioned reward is defined as (for continuous states):*
>
> $$r(o_t^{(1:n)}, a_t^{(1:n)}) = P(m_g(o_{t+1}^{(1)}) = g \mid o_t^{(1:n)}, a_t^{(1:n)}),$$
>
> *with the overall objective being:*
>
> $$\max_{\pi(a^{(i)}|o^{(i)},g)} \mathbb{E}_{p_g(g),\pi(\tau^{(1:n)}|g)} \left[ \sum_{t=0}^{\infty} \gamma^t r(o_t^{(1:n)}, a_t^{(1:n)}) \right]$$

In order to address the concern that picking a goal space $\mathcal{G}$ and mapping function $m_g$ requires user specification and, therefore, provides the method additional information, we test our method by using uninformative and non-task dependent choices of $\mathcal{G}$ and $m_g$. We run our ablation experiment for the SMAX 2s3z environment. Relative to the other tested benchmarks, MPE Tag FACMAC and MABRAX, the SMAX implementation involves the most nontrivial $m_g$: the mapping computes the sum over enemy healths, as opposed to a simple observation truncation (MABRAX) or translation of task reward into goal (MPE Tag). Thus, we believe the results presented here generalize over to other benchmarks.

We let $\mathcal{G} = \mathcal{O}$ (the full observation space) and let $m_g = I$ (the identity map), and provide the method a single goal, commanding a state where every enemy health is 0 while filling out the remaining values using an arbitrary state from a previously collected trajectory. As seen in Figure 9, ICRL achieves non-trivial performance on this task—in fact, our method performs even better than it did with a human-selected choice of $m_g$ that isolates the enemy's health. This shows that ICRL is robust to the choice of $m_g$ in a complex cooperative MARL benchmark and, in fact, may not need this specification at all to complete challenging tasks.

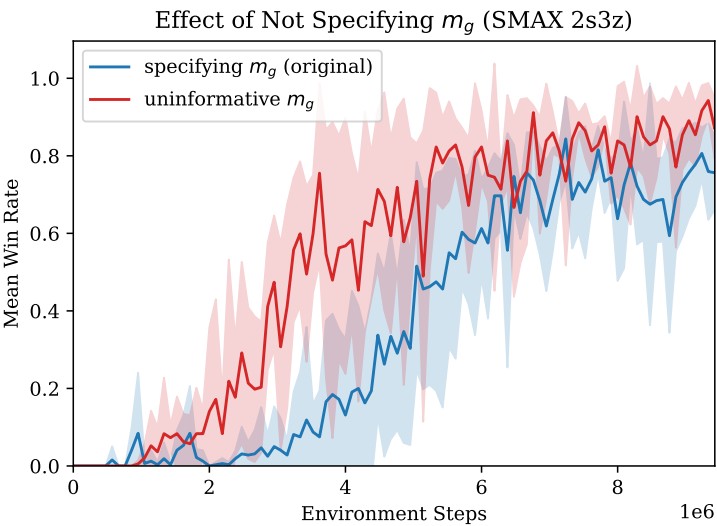

Figure 9: Specifying $m_g$ is not necessary for good performance. $\pm 1\sigma$ error bars.

One hypothesis for why we observe these results is that the full observation provides additional contextual information that may be relevant to goal-reaching. While a manually-designed $m_g$ can help reduce dimensionality and focus learning on intuitive task-relevant observation features, it may inadvertently remove information that is useful in learning more complex environmental dynamics and coordination strategies.

Note that specifying $m_g$ naturally reflects that the goal is a *property* of observations, rather than a specific observation. For instance, although ICRL only receives a single example of the goal in this SMAX experiment, that example still overspecifies the goal: the real objective is to eliminate enemies, not to eliminate enemies *and* reach a randomly-specified state with goal-irrelevant observations. Thus, although the results in Figure 9 suggest $m_g$ is not necessary for ICRL's performance, we keep the goal-mapping function so that our formulation aligns with the nature of real tasks.

### D.3  PROBABILITY OF IMPROVEMENT

In order to determine the statistical significance of our results, we use the probability of improvement, which is the chance that, in a randomly selected environment, a given algorithm performs better than another (Agarwal et al., 2022). We compute the probability of improvement for ICRL compared to MAPPO over all SMAX environments using the max win rate across a training run as the measure of performance, and obtain an estimate of 0.94 with a 95% bootstrapped confidence interval of $[0.85, 0.99]$. When using the final episode win rate instead, we obtain an estimate of 0.86 with a 95% bootstrapped confidence interval of $[0.73, 0.95]$. In both cases, the probability of improvement compared to IPPO is 1.0.

### D.4  DO AGENTS SPECIALIZE?

Given that all agents use identical policy network architectures with unit type as the only distinguishing observation feature, one might expect uniform policy learning across different unit types. Yet prior research demonstrates that role specialization improves performance in cooperative multi-agent tasks (Samvelyan et al., 2019; Witt et al., 2020; Yu et al., 2022). Does our method learn to specialize?

To investigate this, we conduct an ablation study where we systematically remove unit type information from agent observations. We compare three conditions: agents observing both their own and teammates' unit types (baseline), agents observing only teammates' unit types, and agents observing no unit type information. If ICRL relies on unit type specialization, we expect performance to degrade as this information is removed. Figure 10 demonstrates this expected performance decline on 2s3z and the SMACv2 5-unit and 10-unit environments (these are the only environments that include heterogeneous unit types). Interestingly, there is a significantly smaller drop in performance when removing an agent's own unit type from its observation, implying that information about the distribution of unit types across teammates is more relevant to task performance. The 0% win rate on 2s3z when removing all unit type information may seem counterintuitive, especially

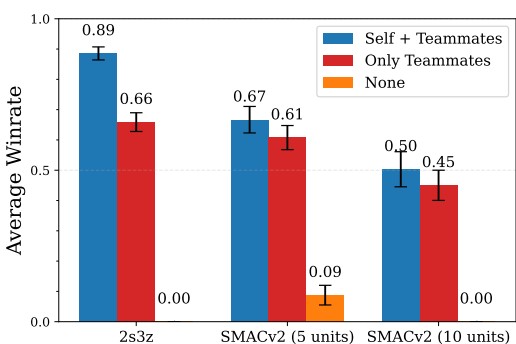

Figure 10: **Agents specialize in SMAC, using information about their player type to make decisions.** Win rates decline as unit type information is progressively removed from agent observations: baseline (self + teammate types), partial ablation (teammate types only), and full ablation (no type information).

since the teammate unit type distribution does not provide extra information beyond the agent's unit type in this environment. We hypothesize that this outcome arises because the policy network was only exposed to a single team unit type configuration during training, and thus does not behave as expected under altered inputs.

## E  DETAILS ON SAMPLING SCHEME

---

**Restatement** (Actor Objective). *The Independent CRL actor objective is as follows:*

$$\max_{\pi} \mathbb{E}_{\substack{o_t^{(i)},g^{(i)}\sim\mathcal{B} \\ a_t^{(i)}\sim\pi(a_t^{(i)}|o_t^{(i)},g^{(i)})}} [-\|\phi(o_t^{(i)},a_t^{(i)}) - \psi(g^{(i)})\|_2]. \tag{9}$$

*where observations and future goals are sampled from the trajectories of the same randomly-chosen agent.*

---

While the actor objective in Eq. 9 samples local observations, actions, and goals as opposed to multi-agent observations, actions, and goals, we can still use this objective as a proxy for the full CRL actor objective.

Concretely, we show here that a modification of Eq. 9 gives a lower bound on the full contrastive actor objective for a given buffer of multi-agent trajectories $\mathcal{B}$, where the sampled goal is a function of all agent observations. The following is a generic statement for multi-agent replay buffers of the form $\mathcal{B} = \{\tau^{(1:N)}\}$, where trajectories are defined over the full multi-agent observation space $\mathcal{O}^{(1:N)} = \mathcal{O}^N$ and full action space $\mathcal{A}^{(1:N)} = \mathcal{A}^N$.

**Assumption E.1** (Actor independence). The joint policy $\pi$ takes the form

$$\pi(a_t^{(1:N)} \mid o_t^{(1:N)}, g) = \prod_i \pi(a_t^{(i)} \mid o_t^{(i)}, g).$$

In words, each individual agent executes actions based on local observations.

**Assumption E.2** (Agent index). Agent index $i$ is included in observation $o^{(i)}$, as is true in empirics.

**Lemma E.3.** *Let a modified version of the Independent CRL actor objective sample goals $g$ that are a function of all agents' collective observations $o^{(1:N)}$:*

$$J_{ICRL, mod}^{\pi} \triangleq \max_{\pi} \mathbb{E}_{\substack{o_t^{(i)},g\sim\mathcal{B} \\ a_t^{(i)}\sim\pi(a_t^{(i)}|o_t^{(i)},g)}} [-\|\phi(o_t^{(i)},a_t^{(i)}) - \psi(g)\|_2]. \tag{10}$$

*where changes to Independent CRL (ICRL) are highlighted in teal.*

*We show that the modified ICRL is a lower bound on the full contrastive actor objective Eq. 11:*

$$J_{full}^{\pi} \triangleq \max_{\pi} \mathbb{E}_{\substack{o_t^{(1:N)},g\sim\mathcal{B} \\ a_t^{(1:N)}\sim\pi(a_t^{(1:N)}|o_t^{(1:N)},g)}} [-\|\phi_{full}(o_t^{(1:N)},a_t^{(1:N)}) - \psi_{full}(g)\|_2]. \tag{11}$$

*where $\phi_{full}$ and $\psi_{full}$ denote the representations learned when sampling concatenated observations $o^{(1:N)}$ and goals $g$.*

*Proof.* At convergence of the full InfoNCE loss, the critic $f_{full}(o^{(1:N)}, a^{(1:N)}, g)$ captures the temporal correlations between *multi-agent* observations, actions, and goals:

$$f_{full}^*(o^{(1:N)}, a^{(1:N)}, g^{(1:N)}) = -\|\phi_{full}^*(o_t^{(1:N)}, a_t^{(1:N)}) - \psi_{full}^*(g)\|_2 \tag{12}$$

$$= \log \frac{\rho^{full}(g \mid o_t^{(1:N)}, a_t^{(1:N)})}{\rho^{full}(g)}. \tag{13}$$

To understand Eq. 11 as a mutual information, we can reformulate this objective in terms of the discounted goal-occupancy measure $\rho_{\gamma}^{full}(o^{(1:N)}, a^{(1:N)}, g)$ induced by the individual policy $\pi(a^{(i)} \mid o^{(i)}, g)$. The $\gamma$ subscript is dropped for simplicity.

Let $G, O^{(1:N)}$, and $A^{(1:N)}$ denote the random variables describing samples $(g, o^{(1:N)}) \sim \mathcal{B}$ and $a^{(1:N)} \sim \pi(\cdot \mid o_t^{(1:N)}, g)$. The full actor objective simplifies to a mutual information:

$$J_{\text{full}}^\pi \approx \mathbb{E}_{\substack{o_t^{(1:N)}, g \sim \mathcal{B} \\ a_t^{(1:N)} \sim \pi(\cdot | o_t^{(1:N)}, g)}} \left[ \log \frac{\rho^{\text{full}}(g \mid o_t^{(1:N)}, a_t^{(1:N)})}{\rho^{\text{full}}(g)} \right] \tag{14}$$

$$= \mathbb{E}_{\substack{o_t^{(1:N)}, g \sim \mathcal{B} \\ a_t^{(1)} \sim \pi(\cdot | o_t^{(1)}, g) \dots}} \left[ \log \frac{\rho^{\text{full}}(g \mid o_t^{(1:N)}, a_t^{(1:N)})}{\rho^{\text{full}}(g)} \right] \qquad \text{(actor independence)}$$

$$= I(G; O^{(1:N)}, A^{(1:N)}). \tag{15}$$

We can additionally define $O^{(I)}$ and $A^{(I)}$ as random variables describing samples $o^{(i)} \sim \mathcal{B}$ and $a^{(i)} \sim \pi(\cdot \mid o_t^{(i)}, g)$. The samples $(o^{(i)}, g)$ are drawn from the mixed discounted state occupancy measure $\rho^{\text{mix}}(o^{(i)}, g)$ with random variable $(I)$ denoting the (uniformly sampled) agent index. Then, the Independent CRL actor objective can be written as

$$J_{\text{ICRL, mod}}^\pi \approx \mathbb{E}_i \mathbb{E}_{\substack{o_t^{(i)}, g \sim \rho^{\text{mix}}(o,g) \\ a_t^{(i)} \sim \pi(\cdot | o_t^{(i)}, g)}} \left[ \log \frac{\rho^{\text{mix}}(g \mid o_t^{(i)}, a_t^{(i)})}{\rho^{\text{mix}}(g)} \right] \tag{16}$$

$$= I(G; O^{(I)}, A^{(I)}). \tag{17}$$

The LB follows from basic information theory inequalities:

$$J_{\text{full}}^\pi \approx I(G; O^{(1:N)}, A^{(1:N)}) \tag{18}$$

$$\geq \frac{1}{N} \sum_i I(G; O^{(i)}, A^{(i)}) \qquad \text{(DPI)}$$

$$= I(G; O^{(I)}, A^{(I)} \mid I) \qquad \text{(definition)}$$

$$= I(G; O^{(I)}, A^{(I)}, I) - \underbrace{I(G; I)}_{=0, \ I \text{ sampled uniformly at random}} \qquad \text{(chain rule)}$$

$$= I(G; O^{(I)}, A^{(I)}) \qquad \text{(Assump. E.2)}$$

$$\approx J_{\text{ICRL, mod}}^\pi \tag{19}$$

where both $\pi$ on the LHS and RHS are functions of single observations and goals by the independence assumption. $\qquad \square$

Thus, the learned policy maximizes a mutual information that lower bounds the true desired mutual information. One cannot bound any distance between the learned policies without additional assumptions.

A final caveat is that ICRL does not directly use the collective goal in critic or actor sampling. Rather, ICRL samples agent-specific goals $g^{(i)}$ as a proxy for the collective goal $g$. While this leads to weaker theoretical motivation for the method, this sampling scheme works well empirically in several challenging collaborative MARL tasks (see Section 5).

