# OpenReview forum: "Self-Supervised Goal-Reaching Results in Multi-Agent Cooperation and Exploration"
_ICLR.cc/2026/Conference — Submitted to ICLR 2026_

### Official Review · Reviewer_n7bM · 2025-11-01

**Soundness:** 3
**Presentation:** 4
**Contribution:** 3
**Rating:** 6
**Confidence:** 4

**Summary:**

The paper proposes Independent Contrastive Reinforcement Learning (ICRL), a self-supervised independent RL learning algorithm that enables learning policies in multi-agent settings by specifying a goal to reach, without having to craft a timestep-based reward function. Namely, the reward is defined automatically as the likelihood of reaching the goal. To achieve this the authors use contrastive learning to train a critic that measures how close a state-aciton pair os to the goal in a learned embedding space. The algorithm is evaluated on cooperative and competitive benchmarks such as Starcarft and a 4-legged robot control. ICLR outperforms strong baselines like IPPO or MAPPO in these sparse reward settings.

**Strengths:**

The paper offers an elegant approach to tackle multi-agent RL problems as a self-supevised goal reaching problem, building on top of ideas from goal-conditioned RL and independent policy learning. The paper is enjoyable to read and well-explained. The results are interesting, showing emergent learning without requiring much exploration signal and without needing to define a reward function. Experiments cover several good environments and compare against strong baselines such as PPO. The motivation to remove dependence on hand-tuned rewards is also well justified.

**Weaknesses:**

- The approach assumes that meaningful goals can be defined, which mgiht not be straightforward in realistic tasks. This is briefly mentioned by authors in the limitations but no suggestions to address this are proposed.
- Empirical results are strong but could benefit from benchmarking on more environments, running ablations to analyze which part of the algorithm leads to the most improvements, and more analysis to understand why the algorithm performs well with such sparse rewards (though the authors provide a qualitative discussion).
- Providing more experimental details about how baselines such as MAPPO and IPPO were calibrated (hyperparameter tuning budget) would help to assess the fairness of the comparison.

**Questions:**

- In the experiments done here do all agents share the same goal?
- Can the authors explain how the standard vs unsupervised approaches differ on a simple problem, such as going from point A to point B? The standard reward would be the negative L2 distance between the agent's position and the goal. In this simple case how would the two approaches compare, and given that the goal conditioned approach would lead to a very sparse reward, do the authors have any intuition for what would the embedding functions learn in order to perform well on this task?
- Can the authors give example of how goals could be defined for more complex realistic RL tasks, for example in robotics or autonomous driving?
- The plots compare performance vs steps. I wonder if runtime is also comparable between IPPO/MAPPO and ICRL?

---

> ### Author Response · Authors · 2025-11-25
>
> We thank the reviewer for taking the time to review our manuscript. So far, we have done the following improvements to address the main concerns:
> 1. Address concerns about goal specification.
> 2. Point to emergent exploration as reason for high performance with sparse rewards.
> 3. Clarify hyperparameter tuning for our method and baselines.
>
> > The approach assumes that meaningful goals can be defined, which might not be straightforward in realistic tasks.
>
> The focus of our paper is on goal-reaching tasks. Nonetheless, we believe that many tasks can be roughly cast as goal-reaching. For example, a foraging task could be formulated as trying to collect (say) 10 trees; it also might be specified as collecting more trees than another player (i.e., the goal is a binary indicator indicating which player has collected more trees). We are currently exploring setting goals dynamically—for example, as current score + k—to avoid needing an arbitrarily large value.
>
> We note that the rest of the JAXMARL [1] benchmark tasks,  none of which were explicitly designed for goal-reaching, can all be roughly cast as goal-reaching:
>
> | Task    | Goal |
> | -------- | ------- |
> | Simple World Comm (good agent) | set vector <food eaten, distance to observed predators> to <K, K>  |
> | Simple World Comm (adversary) | same as MPE Tag (see paper section 6.1)  |
> | Overcooked | set number of pending recipes to zero OR number of soups done to K |
> | Hanabi | form five consecutively ordered stacks (explicit goal of the game) |
> | JaxNav | set self’s position to the destination |
> | STORM | set payoff to K |
> | Coin Game | set payoff to K |
> | Switch Riddle | set payoff to K |
>
> > benchmarking on more environments, running ablations to analyze which part of the algorithm leads to the most improvements, and more analysis to understand why the algorithm performs well with such sparse rewards
>
> We aimed to focus on the most challenging MARL tasks that covered a range of orthogonal skills (e.g. coordination in MPE Tag, continuous control in Ant, and long-horizon coordination in SMAX). Since SMAC is a commonly-used benchmark, we opted to focus on this as a benchmark against baselines. We would like to point to a recent paper [2], which further explains emergent exploration, the mechanism we believe leads to high performance well with such sparse rewards.
>
> > details about how baselines such as MAPPO and IPPO were calibrated (hyperparameter tuning budget)
>
> To the best of our knowledge, the hyperparameters from the JAXMARL paper are tuned to each task (see Appendix G of the JAXMARL paper [1], where each hyperparameter is taken either from the original paper or new hyperparameters specifically for the JAX task if performance is better). Furthermore, performance of JAXMARL is “thoroughly benchmark[ed]” on correctness to ensure “equivalent agent performance” [1]. Note again that we did not hyperparameter tune our method at all, and highlight that our results are for the sparse-reward setting.
>
> Kind regards,
>
> –Authors.
>
> [1] Rutherford, A., Ellis, B., Gallici, M., Cook, J., Lupu, A., Ingvarsson, G., Willi, T., Hammond, R., Khan, A., Witt, C. S. d., Souly, A., Bandyopadhyay, S., Samvelyan, M., Jiang, M., Lange, R. T., Whiteson, S., Lacerda, B., Hawes, N., Rocktaschel, T., Lu, C., and Foerster, J. N. (2024). JaxMARL: Multi-Agent RL Environments and Algorithms in JAX. arXiv:2311.10090 [cs].
>
> [2] Mahsa Bastankhah, Grace Liu, Dilip Arumugam, Thomas L. Griffiths, & Benjamin Eysenbach. (2025). Demystifying the Mechanisms Behind Emergent Exploration in Goal-conditioned RL.

---

### Official Review · Reviewer_AXSW · 2025-11-03

**Soundness:** 2
**Presentation:** 2
**Contribution:** 2
**Rating:** 4
**Confidence:** 3

**Summary:**

This paper introduces Independent Contrastive Reinforcement Learning (ICRL) to address coordination and exploration in multi-agent systems facing sparse feedback signals. ICRL reframes the cooperative Multi-Agent RL (MARL) problem as a self-supervised goal-reaching task, where users specify the objective via a single goal state rather than complex reward functions. ICRL is an actor-critic algorithm that combines insights from single-agent Contrastive RL (CRL) and Independent PPO (IPPO). The critic learns representations using the symmetric InfoNCE loss to parameterize a mixed discounted state occupancy measure, which intuitively captures goal-directed temporal correlations.
Empirically, ICRL demonstrates superior performance and robustness across long-horizon, sparse-reward benchmarks, including multiple difficult SMAX environments. On four SMAX maps, ICRL is the only method to achieve a non-zero win rate compared to IPPO and MAPPO, both using the same sparse reward. Crucially, ICRL achieves high performance without explicit mechanisms for exploration or subgoals, displaying complex emergent cooperation, like micro-management and focus-fire in StarCraft. The paper argues that goal-conditioned contrastive learning makes the multi-agent goal-reaching problem tractable.

**Strengths:**

- Originality

The primary originality lies in successfully extending self-supervised, representation-based goal-conditioned RL (GCRL) to the challenging domain of decentralized Multi-Agent RL (MARL). Prior approaches tackling sparse rewards in MARL predominantly rely on explicit mechanisms like intrinsic motivation, domain-guided search, or hierarchical generation of subgoals (e.g., MASER, CMAE, LAIES). ICRL avoids all these components, learning goal-directed behavior and exploration in a fully end-to-end manner. The formulation casts the multi-agent objective entirely as maximizing the likelihood of visiting a goal state specified by a single observation, which is highly appealing for practitioners burdened by reward engineering.
The paper combines Independent Learning (IL) principles (decentralized policies with parameter sharing, following IPPO) with the Contrastive RL critic design. This combination leads to a novel mixed critic objective $ρ_γ^{mix}$ that averages goal occupancy measures across agents. Although rooted in single-agent CRL, applying this representation learning approach to the partial observability and coordination challenges inherent in multi-agent systems represents a significant methodological step. Furthermore, the empirical finding that factoring a single-agent task (Ant control) into a multi-agent one can accelerate initial learning by effectively restricting the hypothesis space is a counterintuitive and original contribution to understanding MARL structure.


- Quality

The authors perform fair comparison by providing all methods (ICRL, IPPO, MAPPO, MASER) with the exact same sparse 0/1 reward signal (+1 upon winning/reaching the goal, 0 otherwise). ICRL’s performance gap relative to strong baselines on the SMAX benchmark is massive and highly convincing. For instance, on the 2s3z, 6h v 8z, 8m, and 3s v 5z settings, ICRL achieved success rates above 84%, while IPPO and MAPPO achieved 0%. This suggests that ICRL’s goal-conditioned representation learning effectively solves the exploration difficulty inherent in these long-horizon tasks, where standard PPO variants fail to witness a single successful trial.
The head-to-head comparison with MASER, a specialized SOTA sparse-reward MARL algorithm, further strengthens the paper's claims, showing ICRL achieving a 60% win rate on 2s3z by step 5M, whereas MASER’s win rate was negligible. The results are also supported by statistical evidence using the probability of improvement, which shows ICRL improves upon MAPPO 94% of the time across SMAX environments. The experiments cover complex continuous control (Multi-Agent Ant) and include detailed ablations on agent specialization and the robustness of goal mapping $m_g$.


- Clarity

The paper is well-written, clearly defining the shift from standard MARL (Eq. 1) to the goal-reaching objective (Eq. 6). The algorithm (Independent CRL, Alg. 1) is presented clearly as an alternating actor-critic approach. Through visualizations of training runs on 2s3z, the authors show ICRL agents learning complex micromanagement skills like kiting and focus-fire before observing a single success, while IPPO policies remain random.


- Significance

The work validates multi-agent goal-reaching as a feasible paradigm, potentially allowing future MARL development to circumvent the costly and difficult process of reward engineering. The results fundamentally challenge the necessity of hierarchical decomposition or explicit intrinsic motivation mechanisms for long-horizon sparse-reward MARL, positioning ICRL as a conceptually simpler and more effective alternative than prior methods like MASER.

**Weaknesses:**

- Originality

The methodological originality is primarily compositional. The work is a direct hybridization of existing established concepts: Independent Learning (IL) from MARL literature (e.g., IPPO) and Contrastive RL (CRL) from single-agent GCRL literature. The authors explicitly state they combine "insights from recent work on goal reaching in single-agent settings" and independent learning for MARL. The core innovation is demonstrating that this specific composition works dramatically better in the sparse MARL context than prior MARL methods.
Furthermore, the crucial theoretical links rely on approximations or assumptions. For instance, the critic formulation assumes the overall goal g can be approximated as a function of local observations $o^{(i)}$ for the mixed critic derivation. Appendix E, which aims to provide theoretical grounding for the actor objective, proves a lower bound using a modified ICRL objective ($J_π^{ICRL,mod}$) where goals are collective observations, but then immediately notes that the actual implemented ICRL objective uses agent-specific goals ($g^{(i)}$), leading to "weaker theoretical motivation". Relying on empirical success despite admitted weak theoretical motivation slightly diminishes the originality of the structural design choices made for the multi-agent adaptation.

- Quality

The most notable weakness in quality stems from the selection and performance of the baselines (IPPO and MAPPO). Achieving a 0.00 win rate across four out of five complex SMAX environments with standard sparse 0/1 rewards suggests that these baselines may not represent the strongest possible PPO implementations for these specific sparse settings, despite PPO's overall robustness claims in MARL. While the authors compare against MASER, a strong sparse-reward baseline, this comparison is limited to only the 2s3z environment. To robustly claim superiority over all hierarchical or explicit exploration methods, a wider comparative study against MASER or other state-of-the-art sparse MARL techniques (e.g., CMAE or LAIES, which are mentioned but not run against ICRL) would be needed.
For the continuous control Ant task, the result that ICRL achieves faster initial learning but levels off while single-agent CRL continues improving suggests a fundamental limitation of the Independent Learning framework: a trade-off between reduced variance (fast learning) and increased bias (lower asymptotic performance). This trade-off needs deeper systematic investigation across more than one benchmark to fully understand the generalization and asymptotic limits of ICRL, especially compared to centralized goal-conditioned approaches.


- Clarity

The purpose, derivation, and sensitivity of the coefficient 0.01 in Eq. 7 are not discussed in the text, which obscures a part of the critic objective.

Second, the discrepancy between the theoretical derivation in Appendix E and the implemented algorithm (as discussed above) needs resolution. The authors should explicitly justify why sampling agent-specific future goals $g^{(i)}$ (the implemented choice) works well empirically, even if it lacks the theoretical bound proved for the modified objective using collective goals g.

Third, the handling of discrete actions involves parameterizing the critic using the soft actor output instead of the sampled discrete action. The justification provided is that this "slightly speeds up learning". A more detailed explanation of why the continuous relaxation of the action (the soft output) is a better feature input for the contrastive representation learning, which is fundamentally about temporal correlation, is necessary for full clarity and technical rigor, especially given that this is an "unconventional choice".


- Significance

The significance is somewhat conditional on the practical definition of the goal. The authors note the limitations: it may not always be clear how to specify certain tasks as goals. Furthermore, the surprising finding in Appendix D.2 that using an "uninformative" goal mapping $m_g$ (the identity map G=O) can sometimes lead to better performance than a human-selected $m_g$ complicates the core utility proposition: that goal-reaching simplifies task specification. If practitioners must default to using the full observation space as the goal (identity map) because manually defined goal features inadvertently remove useful contextual information, then the claimed benefit of defining the goal space G and mapping $m_g$ is weakened.

**Questions:**

1. Given the extremely low performance of IPPO and MAPPO (0.00 win rates) when using the sparse 0/1 reward signal on challenging SMAX environments (e.g., 2s3z, 6h v 8z, 8m, 3s v 5z), have the authors tested whether incorporating minimal, standard exploration bonuses or noise injection (not goal-specific or intrinsic motivation) into these baselines could significantly close the performance gap with ICRL?

2. The implementation of ICRL samples agent-specific achieved future goals $g^{(i)}$ (Eq. 9), yet the theoretical lower bound relies on a modified objective sampling the collective goal g (Eq. 10). Since the authors note the implemented choice has "weaker theoretical motivation", can they provide a detailed empirical analysis or justification regarding why the agent-specific goal sampling scheme was chosen over the more theoretically grounded collective goal scheme?

3. The symmetric InfoNCE loss includes a regularization term 0.01⋅R(ϕ,ψ). Can the authors clarify the theoretical purpose of this specific regularization and discuss the sensitivity of ICRL's performance to the coefficient 0.01?

4. The Multi-Agent Ant experiment shows ICRL learning much faster initially than single-agent CRL, but then CRL eventually surpasses ICRL. Does this pattern of faster initial learning but lower asymptotic performance hold across other factored continuous control benchmarks? If so, what design modifications (beyond increasing the sample budget) could be incorporated into ICRL to mitigate the inherent bias from the independent learning factorization?

---

> ### Author Response · Authors · 2025-11-25
> **Response (part 1)**
>
> We thank the reviewer for taking the time to review our manuscript. So far, we have done the following improvements to address the main concerns:
>
> - LAGMA (2s3z) and MAPPO noise injection (2s3z) experiments
> - Clarify the handling of discrete actions using the Gumbel Softmax trick.
> - Clarify the role of theoretical statements.
> - Address points in originality, clarity, and significance
>
> Does the following address the reviewer’s concerns and questions?
>
> > originality of the structural design choices made for the multi-agent adaptation.
>
> Yes, our method can be roughly summarized as applying CRL independently to each agent. However, we highlight that our key contribution is clear evidence that a simple, lightweight algorithm that specifies a single goal during rollout with no prior examples can lead to emergent collaborative behavior in complex environments, despite the sparsity of the problem-setting. To the best of our knowledge, we are not aware of any prior work that proposes this method. Nonetheless, we find that it can outperform prior methods by 3x on StarCraft tasks, often being the only method to see a single success (in 4 out of 6 SMAX tasks) in this extremely sparse reward setting.
>
> > discrepancy between the theoretical derivation in Appendix E and the implemented algorithm
>
> We acknowledge the discrepancy. As the reviewer pointed out, we already made note of the discrepancy in the paper itself, and have added additional clarifications within the main body and Appendix E. However, it is not uncommon for ML methods (particularly independent MARL) to have discrepancies between theory and practice. For example, the IPPO paper [1] contains less theoretical justification than what we have here and has been highly influential in showing that effective MARL can be achieved with independent learning. Similarly, we show that independent reward-free learning can also lead to effective MARL, which is a further simplification of the independent learning setting.
>
> In addition, the discrepancy between theoretical motivation and implementation is (1) effectively closed in the experiments we present given our definition of $m_g$ and (2) points to future work that can close this gap.
>
> (1): in our experiments, the m_g^{(i)}defined for individual agents is typically similar to the overall m_g, except we arbitrarily fill out unobserved features of the input. For example, in MPE Tag, distances for unobserved entities are set to infinity.
>
> In information theory terms, $I(G; O^{(I)}, A^{(I)}) \approx I(G^{(I)} ; O^{(I)}, A^{(I)})$ for the empirically-tested mappings.
>
> (2): The theoretical approach simply implies that future goal labeling should take a function over all observations as inputs rather than individual observations (which, due to (1), should not significantly change conclusions of the paper). This is very  similar to the current formulation presented in Algorithm 1 and can be tested empirically in future work.
>
> However, we believe that the results, thorough ablations, precedent from prior highly-influential works in independent MARL like IPPO, and experiments running/ran during the rebuttal period to clarify reviewer concerns already communicate our main contribution: that independent reward-free learning can lead to effective, robust, and scalable MARL for challenging coordination problems.
>
> > If practitioners must default to using the full observation space as the goal (identity map) because manually defined goal features inadvertently remove useful contextual information, then the claimed benefit of defining the goal space G and mapping is weakened.
>
> The purpose of Appendix D.2 is to show that the method is not overly-sensitive to the definition of $m_g$ in the most extreme case. Critically, in the SMAX problem, we have the ability to directly relabel our data to reflect a goal state with dead enemies. This is to make clear that the method is robust: even if the goal is overly-specified, the method can still work to achieve goals.
>
> However, the ability to set the goal state as an achievable pre-image of a goal (as is done here with the identity mapping) is non-trivial and can be impossible. Consider the extreme limit where the goal function is decided by an oracle for which we have no weights (or a function that is very computationally challenging to invert). In this case, it’s impossible to condition the goal as a full observation-space vector – there is no way we can invert this function without additional assumptions-- but it’s entirely possible to train on this signal with the oracle mapping function...
>
> In summary, the experiment presented in Appendix D.2 is to show that the method is robust, but setting $m_g$ to the identity map is not a generalizable method across goal-reaching tasks: nontrivial $m_g$ mappings are still necessary for the set of problems where finding goal pre-images is challenging/impossible in the oracle case (without additional assumptions, training of generative models, etc.).

---

> ### Author Response · Authors · 2025-11-25
> **Response (part 2)**
>
> > wider comparative study against MASER or other state-of-the-art sparse MARL techniques
>
> We are currently running LAGMA [2], which makes a critical contribution in goal-conditioned MARL. In our paper, we choose to focus on “super-sparse” binary rewards (1 for a win and 0 otherwise), which we believe is also very useful. In 2s3z, LAGMA achieves 0 reward for all training timesteps up to 1M.
>
> > For the continuous control Ant task…This [bias-variance] trade-off needs deeper systematic investigation … especially compared to centralized goal-conditioned approaches.
>
> We agree the Ant curves show faster initial improvement for ICRL with a plateau later on. Our other experiments suggest this is not a fundamental limitation of Independent Learning, but rather reflects task-specific factors (e.g., the decomposition of Ant’s locomotion problem and representation capacity of per-limb policies). In other benchmarks, ICRL continues improving asymptotically. We will clarify this and discuss ongoing experiments adding limited communication for this task, which we believe will show this asymptotic gap is based on task-specific factors, not fundamental factors.
>
> Note that the primary finding of this research is that this problem setting is tractable with a self-supervised contrastive approach – independent learning is not a fundamental requirement of our formulation. We can just as easily employ a monolithic critic and take advantage of the benefits of lowered bias.
>
> >The purpose, derivation, and sensitivity of the coefficient 0.01 in Eq. 7 are not discussed in the text, which obscures a part of the critic objective.
>
> The choice of using logsumexp regularization for the critic comes from prior work which found it helps the classifier to more accurately estimate probabilities [3]. 0.01 is an arbitrary constant within the correct order of magnitude. Any other reasonable constant could be chosen without hurting performance significantly.
>
> > more detailed explanation of why the continuous relaxation of the action (the soft output) is a better feature input for the contrastive representation learning…is necessary.
>
> The continuous relaxation of the action (Gumbel-softmax trick [5]) is a standard way to train actor-critic reinforcement learning algorithms under discrete actions. It is not possible to take gradients of categorical samples; however, these categorical samples are what the environment sees. Thus, the critic is typically given discrete actions during forward pass, but the gradients are taken as if soft actions were used. This means that the gradient is being approximated. However, by using soft actions during the forward pass as well as the backward pass, we avoid any such approximation. This is the only adjustment we make. We suspect that it results in better performance for our method because the representations learned by the critic are essential to early-stage learning, and so it is crucial that these be expressive and accurate.
>
> >  minimal, standard exploration bonuses or noise injection…into these baselines could significantly close the performance gap with ICRL?
>
> We ran experiments with MAPPO (the most performant baseline on SMAX) on the 2s3z environment with several standard exploration mechanisms. We tried the following:
> - added action space Gaussian noise
> - action entropy term in actor loss
> - parameter space noise [4]
>
> We tried these modifications with 5 random seeds and swept over the relevant hyperparameters (standard deviation of Gaussian noise, entropy coefficient). We found that zero successes were achieved even with these modifications.
>
> Kind regards,
> –Authors.
>
> [1] C. Schroeder de Witt, et al. (2020). Is Independent Learning All You Need in the StarCraft Multi-Agent Challenge?.
>
> [2] Hyungho Na, & Il-chul Moon. (2024). LAGMA: LAtent Goal-guided Multi-Agent Reinforcement Learning.
>
> [3] Eysenbach, B., et al. (2023). Contrastive Learning as Goal-Conditioned Reinforcement Learning.
>
> [4] Plappert, M., et al. (2017). Parameter Space Noise for Exploration.
>
> [5] Jang, E., et al. (2016). Categorical reparameterization with gumbel-softmax.

---

### Official Review · Reviewer_Tf73 · 2025-11-04

**Soundness:** 2
**Presentation:** 3
**Contribution:** 2
**Rating:** 4
**Confidence:** 4

**Summary:**

This paper presents a study on multi-agent Reinforcement Learning, specifically tackling the difficulty of learning in sparse reward environments. The authors propose an integrated method combining Goal-Conditioned RL and Contrastive Learning. Crucially, the approach leverages contrastive learning to maximize the likelihood of visiting specific goal states. The proposed method was evaluated on a subset of the StarCraft Multi-Agent Challenge (SMAC) and Multi-Agent Particle Environment (MPE) tasks, where experimental results show it  achieves non-zero rewards on several complex, sparse-reward tasks.

**Strengths:**

1. The experimental results show that the proposed method, despite its simplicity, effectively achieved good performance on a challenging subset of tasks within the SMAC benchmark.

2. While some clarification on technical details are needed, the paper is generally well-organized and easy to follow.

3. The reviewer noted that the emergent exploration behavior was engaging. Agents acquired fundamental skills early in training and progressively developed more sophisticated strategies, indicating that goal-conditioned RL has the potential to enhance exploration.

**Weaknesses:**

1. The reviewer has some concerns regarding the paper's core technical contribution. The proposed methods, particularly the goal-conditioned reinforcement learning framework and the use of a contrastive loss, appear to share significant similarities with the existing LAGMA [a] framework. The reviewer attempted to identify other major technical innovations beyond this adaptation but did not find them clearly articulated. To assist readers in appreciating the novel aspects of this work, the reviewer suggests that the authors could strengthen the paper by explicitly delineating the primary technical differences between their proposed method and previous works, especially LAGMA.

2. The experimental section would be more compelling if the authors compared their approach not only with IPPO and MAPPO, but also with goal-conditioned RL methods like LAGMA, especially given LAGMA's reported non-zero rewards in sparse reward settings for SMAX 8m, 2s3z, and 2m1z.


[a] LAGMA: LAtent Goal-guided Multi-Agent Reinforcement Learning, Na et al. ICML 2024

**Questions:**

1. The paper's description of how goals are defined within each task environment is currently unclear. Could the authors please provide a more specific definition of what constitutes a 'goal state,' particularly for the MPE-tag and SMAC tasks?

2. Is the difficulty of the SMAC tasks super-hard?

---

> ### Author Response · Authors · 2025-11-25
>
> We thank the reviewer for taking the time to review our manuscript. So far, we have done the following improvements to address the main concerns:
> 1. Additional experiments: LAGMA (2s3z)
> 2. Delineate several of the major differences between our method and LAGMA
> 3. Clarify experimental details including goal states and task difficulty
>
> > explicitly delineating the primary technical differences between their proposed method and previous works, especially LAGMA.
>
> Although both LAGMA and our method use goals to help guide exploration when rewards are sparse, the difference is that our method uses hindsight relabelling to learn representations instead of a VQ-VAE: this allows there to be a useful representation of the goal even when the agents have never seen a single goal reaching trajectory. On the other hand, LAGMA can only start using goal-guided search once it has experienced a positive reward, until then it is exploring blindly. Our approach allows us to operate in extremely sparse regimes where signal is only given when a success is achieved, much sparser than what LAGMA uses.
>
> Another big difference between the methods is that our method is goal-conditioned, not goal-guided, meaning that we directly learn to achieve specified goal states instead using goals to provide an additional intrinsic reward alongside an extrinsic reward. Finally, we learn to reach a single goal state instead of learning to follow an entire goal reaching reference trajectory. This allows the method to find more diverse paths to the goal during exploration.
>
> > compared their approach not only with IPPO and MAPPO, but also with goal-conditioned RL methods like LAGMA
> We are currently running LAGMA, which makes a critical contribution in goal-conditioned MARL. In our paper, we choose to focus on “super-sparse” binary rewards (1 for a win and 0 otherwise), which we believe is also very useful.
>
> Below are preliminary results we have on the 2s3z environment (binary “super-sparse” rewards):
> | env timestep    | test_battle_won_mean |
> | -------- | ------- |
> | 0 | 0.0  |
> | 0.25 M | 0.0 |
> | 0.5 M | 0.0 |
> | 0.75 M | 0.0 |
> |1 M | 0.0 |
>
> > Could the authors please provide a more specific definition of what constitutes a 'goal state,' particularly for the MPE-tag and SMAC tasks?
>
> Precisely stated, a goal state is the application of $m_g$ on a state. $m_g$ can be thought of as a dimension reduction of the state space into a space of goals. For MPE-tag, $m_g$ maps a state into the distance from the prey to the nearest predator. For SMAC, $m_g$ maps a state into the sum of enemy healths (we also tried identity $m_g$ in an ablation study in the Appendix).
>
> > Is the difficulty of the SMAC tasks super-hard?
>
> Yes, the difficulty of all the SMAC tasks we used are super-hard, precisely because we use a 0/1 sparse reward. This is a very sparse signal and renders the exploration problem very difficult. In addition, we also tested the algorithm on SMACV2 environments with 5 and 10 randomly generated agents: these tasks are known to be super-hard even in the dense reward setting [1]. We also note that papers such as LAGMA use SMAC environments such as 2s3z, 3m, and 6h_v_8z as tests of performance in the sparse reward setting.
>
> Kind regards,
> –Authors.
>
> [1] ​​Benjamin Ellis, Jonathan Cook, Skander Moalla, Mikayel Samvelyan, Mingfei Sun, Anuj Mahajan, Jakob N. Foerster, & Shimon Whiteson. (2023). SMACv2: An Improved Benchmark for Cooperative Multi-Agent Reinforcement Learning.

---

### Official Review · Reviewer_Eavo · 2025-11-08

**Soundness:** 2
**Presentation:** 2
**Contribution:** 2
**Rating:** 4
**Confidence:** 3

**Summary:**

This paper proposes Independent CRL (ICRL), a method that extends contrastive reinforcement learning to multi-agent settings for goal-reaching tasks. The key idea is to frame multi-agent coordination as a goal-conditioned problem where agents maximize the likelihood of reaching a commanded goal state rather than optimizing a scalar reward. Each agent learns independently using shared parameters, with a contrastive critic learning representations that capture temporal relationships between state-action pairs and goals. The authors evaluate their approach on MPE Tag, StarCraft Multi-Agent Challenge (SMAC/SMAX), and multi-agent continuous control tasks, demonstrating superior performance compared to IPPO, MAPPO, and MASER baselines in sparse-reward settings.

**Strengths:**

1. The problem is well-motivated. The goal-conditioned multi-agent framework removes the burden of reward engineering while maintaining clear task specification through a single goal state.

2. The paper demonstrates positive results across diverse environments (MPE Tag, SMAC/SMAX, multi-agent continuous control) with different team sizes and complexity levels. On several SMAC environments (2s3z, 6h_v_8z, 8m, 3s_v_5z), ICRL achieves non-zero win rates while baselines fail completely. The analysis in Section 5.3 provides useful qualitative insights into emergent coordination strategies.

3. Section 5.5 shows that factoring a single-agent problem into a multi-agent one can accelerate learning, which is counterintuitive and potentially valuable.

**Weaknesses:**

### Major Issues

1. **Unclear mechanism for early-stage learning**: The paper states that "g_i are future states encountered after (o_i, a_i)" (line 207), but elsewhere discusses g as a specific goal state. This creates critical ambiguities:
   - When training the critic, are "achieved future goals" any future states in a trajectory, or only states where the final goal is reached? How about when training the policy?
   - If agents never reach the goal state g* during early training, how does the InfoNCE loss (Eq. 7) trained on arbitrary future states g_i provide learning signal toward the specific goal g*?
   - The connection between training on arbitrary achieved goals and transferring to a specific commanded goal g* needs explicit discussion.

   This is particularly important given the paper's claim that the method works in settings where "baselines never observe a single success"—if the goal is never observed, what guides the policy toward it?

2. **Insufficient justification for key assumptions and design choices**:
   - The method assumes the overall goal g can be approximated as a function of local observations o^(i), but provides no principled guidance for when this is applicable. For instance, in SMAC environments, how is the goal state determined to be reached when some enemies are outside all agents' observation ranges?
   - While Lemma E.3 provides a lower bound for a *modified* version of ICRL, the actual algorithm samples agent-specific goals g^(i) rather than collective goals g. The authors acknowledge this gap between theory and practice but don't adequately explain why the empirically-used sampling scheme should work.

3.. **Experimental setup concerns**:
   - The sparse reward formulation (0/1 only at goal achievement) may be particularly disadvantageous for standard RL methods designed for dense rewards. While the authors argue this is "natural," it's quite extreme and may not reflect realistic deployment scenarios.
   - The paper identifies LAGMA (Na & Moon, 2024) as "perhaps the most related work" in Section 2, as both methods address goal-conditioned multi-agent RL in sparse reward settings. However, no experimental comparison with LAGMA is provided.

### Minor Issues

   - Line 118: s_t missing superscript (1:n)
   - Line 132: "Dec-POMDP" should be spelled out on first use
   - Line 170/Equation 5: The probability function should use larger parentheses; the current formatting makes it appear as if there are three separate equalities (A=B=C), causing potential confusion
   - Lines 200-201: "Formally" is used twice in succession; rephrase for better flow
   - Line 212/Equation 7: Uses subscript i for agent ID while the rest of the paper uses superscript (i); notation should be consistent

**Questions:**

see Weaknesses.

---

> ### Author Response · Authors · 2025-11-25
>
> We thank the reviewer for taking the time to review our manuscript. So far, we have done the following improvements to address the main concerns:
> 1. Additional experiments: LAGMA baseline (2s3z)
> 2. Clarify achieved vs commanded goals and why early-stage learning works.
> 3. Address concerns regarding goal specification and sparse rewards
>
> > When training the critic, are "achieved future goals" any future states in a trajectory...?
>
> To clarify, achieved future goals are any future states in a trajectory ($m_g$ applied to these states), when training both the critic and policy. We distinguish between **commanded** goals (where we want agents to get) and **achieved** goals (where they currently are).
>
> The notion of commanded/achieved goals comes from hindsight-experience relabeling [4]. Namely, while we only use the single commanded goal during *rollout* (for which the agent may never have seen a positive example), we *train* our critic and update our actor using *actually achieved “goals”*. This approach allows us to use arbitrary future “goals” as a proxy to learn about our target goals.
>
> This proxy can provide actual learning signals because these arbitrary future “goals” help the agent learn general skills that can be applied to reach the target goal, and these arbitrary future “goals” can provide examples for what not to do, guiding future exploration efforts [3].
>
> > If the goal is never observed, what guides the policy toward it?
>
> We attribute this early-stage learning to emergent exploration, a phenomenon exhibited by many self-supervised RL algorithms—the mechanism behind emergent exploration is still an open problem which is why there is no detailed explanation in our manuscript.
>
> However, concurrent work has made progress toward elucidating this phenomenon [3] and we summarize the key results here: learning a representation space which encodes the environment dynamics allows unsuccessful trajectories to be pruned from future exploration efforts. This allows agents to make progress toward the goal without explicit supervision through negative feedback.
>
> > Insufficient justification for key assumptions and design choices. “The method assumes the overall goal g can be approximated as a function of local observations o^(i), but provides no principled guidance for when this is applicable.”
>
> This assumption that the overall goal $g$ can be approximated from local observations arises from needing decentralized agents to determine whether they are making progress towards their goal. We only need to approximate the goal in this way if we want agents to learn *completely* independently.  We can explicitly test how much information loss learners are robust to for a given environment by training with different observation radii and plotting performance curves. Note that this assumption is *not fundamental* to the method. In fact, we can also easily use global goals to train our method with minimal changes to avoid this limitation (clarified in the manuscript).
>
> > in SMAC environments, how is the goal state determined to be reached when some enemies are outside all agents' observation ranges?
>
> To compute the goal state, we simply set healths to full for enemies outside of an agent’s observation radius. This means that goal states are a function of local observations, and that the achieved goals used during training are *different* from what the actual goal may have been achieved. For example, if all the enemies have died but some enemies are unobserved, then the goal would not be considered achieved.
>
> > Experimental setup concerns
>
> We agree that, in some settings, it may be better to use dense rewards methods. We focus on the “super-sparse” reward (1 for reaching a goal state and 0 otherwise) because it enables new tasks (e.g. cell development or bird flocking). Prior MARL works [1, 2] often compare to baselines not explicitly designed for sparse rewards (e.g. QMIX). Note some environments such as MPE Tag are already effectively 0/1 super-sparse rewards.
>
> As far as we are aware, MASER is a more fair comparison against our method than LAGMA because it demonstrated performance on sparser rewards. We are also now running LAGMA. In our paper, we choose to focus on “super-sparse” binary rewards (1 for a win and 0 otherwise), which we believe is also very useful. All test_battle_won_means are 0 over the course of 1M training steps.
>
> > Minor issues
> Thank you for pointing out these minor errors. We have fixed them in the manuscript.
>
> Kind regards,
>
> –Authors.
>
> [1] Jeewon Jeon, et al. (2022). MASER: Multi-Agent Reinforcement Learning with Subgoals Generated from Experience Replay Buffer.
> [2] Hyungho Na, & Il-chul Moon. (2024). LAGMA: LAtent Goal-guided Multi-Agent Reinforcement Learning.
> [3] Bastankhah, M., et al. (2025). Demystifying the Mechanisms Behind Emergent Exploration in Goal-conditioned RL.
> [4] Andrychowicz, M., et al. (2017). Hindsight experience replay. Advances in neural information processing systems, 30.

---

### Author Response · Authors · 2025-12-03
**Note to AC**

Dear AC,

We thank the AC and reviewers for taking the time to review the manuscript. Below, we summarize the main contributions of the paper, the reviewer feedback, and the changes we’ve made to the paper to incorporate the feedback, which further strengthens the paper.

What is this paper about?
- Our key contribution is clear evidence that even a simple, lightweight algorithm that specifies a single achieved goal during rollout can lead to emergent collaborative behavior in highly complex environments, despite the sparsity of the problem-setting (reward of 1 for winning, 0 otherwise).
- Despite its simplicity, we find our method still outperforms prior methods by 3x on StarCraft tasks, often being the only method to see a single success (in 4 out of 6 SMAX tasks) in this extremely sparse reward setting.
- We also observe emergent exploration—agents learn useful skills even before seeing a single success.

Summary of Reviewer Concerns:

Reviewer Eavo (score 4) primarily wanted clarification about how our method works, including our notion of goals and how early-stage learning is possible despite no successes. They also requested more comparison against sparse reward baselines like LAGMA, a concern that was shared by reviewers Tf73 (score 4) and AXSW (score 4). Reviewer Tf73 was mostly concerned about the novelty of our method, especially as compared to LAGMA. Reviewer AXSW primarily had concerns about the method exposition, including our handling of discrete actions, choice of regularization, and theoretical justification. They also requested we try our baselines with standard exploration ablations. Reviewer n7bM (score 6) had minor questions about goal specification and hyperparameter tuning.

Summary of Changes to the Paper:
- Added LAGMA baseline experiments on 2s3z (with 0/1 sparse rewards). LAGMA achieves **0% peak win rate across 1M training timesteps (vs. 36% peak win rate for our method)**, addressing requests for comparison with more sparse reward methods from reviewers Eavo, Tf73, and AXSW.
- Added MAPPO exploration bonus ablations (action space Gaussian noise, action entropy term in actor loss, parameter space noise) on 2s3z. **We found that zero successes were achieved even with these modifications.** This experiment directly addresses a request for baselines with exploration bonuses from reviewer Tf73.
- Clarified the distinction between commanded and achieved goals via hindsight relabeling, and cited recent work (Bastankhah et al., 2025) explaining emergent exploration mechanisms, addressing concerns about early-stage learning from reviewer Eavo. Clarified goal state computation for partial observability (unobserved enemies assigned full health), answering questions about goal specification from reviewers Eavo and Tf73. Clarified role of Gumbel-softmax for discrete actions and logsumexp regularization coefficient, among other exposition related clarifications, addressing questions raised by reviewer AXSW.
- Delineated the differences between our method and LAGMA in response to concerns from reviewer Tf73.

We would like to flag that, according to Pangram, one review was fully AI-generated and two were moderately/heavily AI-edited. While we have addressed all reviewer concerns, we would like to raise this context for consideration for the AC.

We note that review Tf73 inaccurately stated that our paper has no “major technical contributions beyond an adaptation of LAGMA.”  However, LAGMA bears few similarities to our paper beyond the use of goals. In fact, LAGMA operates in a fundamentally different setting than we propose in our paper—it uses goals as an additional incentive rather than as a direct objective. When we ran LAGMA in our goal-conditioned setting, it was not able to learn.

Kind regards,

–Authors

---

### Meta-Review · Area_Chair_WHus · 2026-01-03

**Summary:**

The submission initially received relatively negative reviews (4446). The main concerns can be summarized into the following key points:
1. The basic assumption about the goal estimation is not convincing.
2. The key technical contributions are limited. It directly combines two established concepts: independent learning from MARL and contrastive RL from single-agent GCRL. The key technical differences between the proposed method and the previous method, LAGMA, are unclear.
3. The compared baselines (IPPO and MAPPO) are not strong enough, some goal-conditioned RL baselines are expected.
4. Some implementation details are unclear.
5. More comprehensive ablations are needed.

In the rebuttal, the authors provide a detailed explanation of the comparisons with LAGMA and additional baseline results. Meanwhile, the authors have added more explanations about some details. Overall, I think the concerns about the novelty still remain, and the presentation needs to be further improved. Thus, I recommend Reject.

**Reviewer Concerns:**

The concerns about comparisons with LAGMA and more implementation details or ablation studies have been addressed. But the concerns about the novelty remain.

**Reviewer Scores:**

I think all reviewers will not change their scores.

---

### Decision · Program_Chairs · 2026-01-26

Reject